

# Aerosol particle size distribution in the stratosphere retrieved from SCIAMACHY limb measurements

Elizaveta Malinina[1], Alexei Rozanov[1], Vladimir Rozanov[1], Patricia Liebing[1*], Heinrich Bovensmann[1], and John P. Burrows[1]

[1]Institute of Environmental Physics (IUP), University of Bremen, Bremen, Germany
[*]now at Leiden Observatory, University of Leiden, Leiden, the Netherlands

*Correspondence to:* Elizaveta Malinina (malininaep@iup.physik.uni-bremen.de)

**Abstract.** Information about aerosols in the Earth's atmosphere is of a great importance in the scientific community. While tropospheric aerosol influences the radiative balance of the troposphere and affects human health, stratospheric aerosol plays an important role in atmospheric chemistry and climate change. In particular, information about the amount and distribution of stratospheric aerosols is required to initialize climate models, as well as validate aerosol microphysics models and investigate geoengineering. In addition, good knowledge of stratospheric aerosol loading is needed to increase the retrieval accuracy for the key trace gases (e.g. ozone or water vapor), when interpreting remote sensing measurements of the scattered solar light. There are several parameters which are commonly used to describe stratospheric aerosols, such as the aerosol extinction coefficient and Ångström coefficient. However, the use of particle size distribution parameters coupled with the aerosol number density is an unambiguous and more optimal approach. In this manuscript we present a new retrieval algorithm to obtain the particle size distribution of the stratospheric aerosol from space borne observations of the scattered solar light in the limb viewing geometry. While the mode radius and width of the aerosol particle size distribution are retrieved, the aerosol particle number density remains unchanged. The latter is justified by a lower sensitivity of the limb-scattering measurements to changes in this parameter. To our knowledge this the first data set providing two parameters of the particle size distribution of the stratospheric aerosol from space borne measurements of the scattered solar light. Generally, the mode radius and absolute distribution width can be retrieved with the uncertainty of less than 20%. The algorithm was successfully applied to the tropical region (20°N-20°S) for 10 years (2002-2012) of SCIAMACHY observations in limb viewing geometry, establishing a unique data set. Analysis of this new climatology for the particle size distribution parameters showed clear increases of the mode radius after the tropical volcanic eruptions, whereas no distinct behaviour of the absolute distribution width could be identified. A tape recorder, which describes the time lag as the perturbation propagates to higher altitudes, was identified for both parameters after the volcanic eruptions. A Quasi Biannual Oscillation (QBO) pattern at upper altitudes (28-32 km) is prominent in the anomalies of the analyzed parameters. A comparison of the aerosol effective radii derived from SCIAMACHY and SAGE II data was performed. The average difference is found to be around 30% at the lower altitudes decreasing with increasing height to almost zero around 30 km. The data sample available for the comparison is, however, relatively small.





## 1 Introduction

Stratospheric aerosols are of a great interest for researchers because of their impact on the climate. The radiative budget of the Earth is altered by stratospheric aerosol, which scatters electromagnetic radiation in the atmosphere. Large amounts of aerosol emitted by the volcanic eruptions significantly change the radiative forcing affecting the global temperatures. For example, the

super-colossal Tambora eruption in 1815 is claimed to cause "the year without summer" in 1816. The more recent colossal eruption of Mount Pinatubo in 1991 caused the surface temperature cooling of few tenth of a Kelvin (Robock and Mao, 1995). Not only catastrophic eruptions influence climate, Solomon et al. (2011) showed, that even smaller volcanic eruptions noticeably impact the global temperature. Another important role of aerosols is their ability to act as condensation nuclei for polar stratospheric clouds (PSC) which provide surface for heterogeneous chlorine activation and denitrification processes,

thus, increasing ozone depletion (Solomon, 1999).

Accurate information about stratospheric aerosols is important for different research fields. Among others, stratospheric aerosol parameters are needed for modelling of the processes related to the stratosphere, including aerosol microphysics models validation. Modellers require stratospheric aerosol climatologies as initial conditions for climate model simulations and predictions, e.g. Solomon et al. (2011); Fyfe et al. (2013). Fyfe et al. (2013) concluded that not only temperature is affected by

the volcanic eruptions, but also precipitation, sea level pressure and wind speed change in response to changing aerosol load. All the above mentioned data is needed not only for improving global climate models, but is also required for the investigation of geoengineering. In addition accurate knowledge of the stratospheric aerosol loading is essential to improve trace gases retrievals from measurements of the scattered solar light (Rozanov et al., 2011; Arosio et al., 2017).

As summarized by Kremser et al. (2016) , stratospheric aerosols are commonly described as spherical droplets composed

of a mixture of the sulfuric acid $H_2SO_4$ and water $H_2O$. They might also comprise meteoric dust as well as other nonsulfate material. The most important sources of stratospheric aerosols are carbonyl sulfide (OCS) and sulfur dioxide ($SO_2$) which are oxidized to sulfuric acid. Sulfur dioxide, which is short lived, is mostly injected directly into the lower stratosphere by large volcanic eruptions, while (OCS), which is long lived in the troposhere, is transported in the tropics from the troposphere by convection. Aerosol composition in the stratosphere is controlled by the Brewer-Dobson Circulation, as well as by chemistry

and microphysics during aerosol formation, growth and removal through sedimentation. Although there have been some proposals that anthropogenic activity results in increase of stratospheric $SO_2$, the current consensus is that stratospheric aerosols precursors have predominant natural sources (volcanic eruptions and biomass burning).

One of the most commonly used characteristics to describe stratospheric aerosols is their extinction coefficient. Although it was employed in many previous studies dealing with space borne measurements in UV-visible-IR spectral range (Ovigneur

et al., 2011; Taha et al., 2011; Ernst, 2013; Brinkhoff et al., 2015; Dörner, 2015; von Savigny et al., 2015; Rieger et al., 2015, 2017), the usage of this parameter has several drawbacks. First the extinction coefficient is wavelength dependent and this dependence is determined by the composition of the aerosol droplets and their particle size distribution. Second, while a direct evaluation of the aerosol extinction can be done for solar occultaion measurements, for limb measurements this parameter is dependent on the assumed particle size distribution parameters. In contrast to previous studies, our retrieval characterizes



stratospheric aerosols by their particle size distribution parameters as it is widely used in the in-situ measurements (Deshler et al., 2003). Most commonly, mode radius, distribution width and the particle number density are used to describe the particle size distribution of stratospheric aerosols. These parameters are, however, challenging to retrieve. Ortherwise some information about the particle size distribution can be obtained from the Ångström coefficient, which is related to the wavelength

dependence of the aerosol extinction coefficient. Providing some information on the size of the aerosol particles, Ångström coefficient cannot, however, be uniquely transformed into the particle size distribution parameters. Generally, different combinations of the mode radius and distribution width may result in the same value of the Ångström coefficient. Furthermore, for each particular particle size distribution the resulting Ångström coefficient usually depends on the wavelength range.

Stratospheric aerosols can be measured in situ from balloons and aircrafts as well as by remote sensing instruments as

ground-based lidars (Deshler et al., 2003). However, these measurements are sparse. Thus, to obtain an understanding of the global behavior of stratospheric aerosols space borne measurements are needed. Measurements of the stratospheric aerosols from space borne instruments started in the late 1970s with SAM (Stratospheric Aerosol Measurement), SAM II, SAGE (Stratospheric Aerosol and Gas Experiment) and SAGE II instruments (Yue et al., 1989). The latter was one of the most distinguished instruments of this era operated from 1984 till 2005 and used solar occultation technique. SAGE II product provides

the aerosol extinction at four wavelengths, and some information on particle size distribution parameters (more detailed in Sec. 5). In the beginning of 2000s several space borne instruments of newer generation using stellar occultation and limb measurement techniques were launched. SCIAMACHY (Scanning Imaging Absorption Spectrometer for Atmospheric CHartographY) was one of these instruments operated on the Envisat satellite from 2002 till 2012. SCIAMACHY had three main measurement modes: nadir, solar occultation and limb scattering. Other instruments on board Envisat providing stratospheric aerosol

information were Global Ozone Monitoring by Occultation of Stars (GOMOS) made measurements using stellar occultation technique and providing aerosol extinction coeffiecient (Vanhellemont et al., 2016) as well as the MIPAS (Michelson Interferometer for Passive Atmospheric Sounding) instrument measuring limb emission spectra (Fischer et al., 2008) and providing vertical profiles of $SO_2$ and sulfate aerosol ($H_2SO_4$) volume densities (Günther et al., 2017). OSIRIS (Optical Spectrograph and InfraRed Imager System) on board Odin satellite, which was launched in 2001 and is still operating, measures scattered

solar light in limb viewing geometry (Llewellyn et al., 2004). Using this data aerosol extinction profiles and Ångström coefficient are retrieved (Bourassa et al., 2012; Rieger et al., 2014). The most recent instrument, which combines nadir and limb scattering measurements, is Ozone Mapping Profiler Suite (OMPS) (Jaross et al., 2014). OMPS was launched by NASA in the end of 2011 and provides data till now, the first retrievals of aerosol extinction from this instruments are presented in several publications (Taha et al., 2011; Arosio et al., 2017; Loughman et al., 2017). Besides, it's important to mention the space-based

lidar CALIOP (Cloud-Aerosol Lidar with Orthogonal Polarization Lidar) providing aerosol extinction coefficient profiles with highest vertical resolution possible for the space borne instruments (Vernier et al., 2011).

This manuscript has the following structure: in Sec. 2 the most essential information about the SCIAMACHY instrument and the retrieval method used in the study is presented. In Sec. 3 the sensitivity of the retrieval algorithm is investigated. Results of the SCIAMACHY data retrieval along with the discussion are presented in Sec. 4. In Sec. 5 retrieval results from

SCIAMACHY are compared to the SAGE II particle size distribution product. Conclusions are given in Sec. 6.



## 2 Instrument and applied algorithm

### 2.1 SCIAMACHY on Envisat

SCIAMACHY was a national contribution to Envisat (European Environmental Satellite) launched by ESA on 1 March 2002 into a sun-synchronous orbit at 800 km altitude with a descending node equator-crossing time of 10:00 a.m. SCIAMACHY

provided data from August 2002 till April 2012, when the connection with the satellite was suddenly lost. Instrument operated in three modes: nadir, limb and solar/lunar occultation, as well it provided daily measurements of the solar irradiance. SCIA-MACHY took measurements in 8 spectral channels, which covered the spectral interval between 214 and 2386 nm. Spectral resolution varies from 0.2 to 1.5 nm depending on the wavelength.

  In this study, measurements taken in limb viewing geometry were used. In this geometry the instrument observed the atmo-

sphere tangentially to the Earth's surface in the altitude range from about 3 km below the surface, i.e., when Earth's surface is still within the field of view of the instrument, and then scanning vertically up to the top of the atmosphere (at about 100 km tangent height). The measurements were performed in 3.3 km tangent height steps with the instantaneous geometrical field of view at tangent point of 2.6 km in vertical direction and 110 km in horizontal direction. A horizontal scan with the total swath of about 960 km was performed during each observation sequence. The horizontal cross-track resolution was mainly deter-

mined by the integration time during the horizontal scan reaching typically about 240 km. For a typical limb measurement, the observed signal integrated by the instrument was readout four times per horizontal scan that resulted in four independent limb radiance profiles obtained during a horizontal scan. The horizontal along-track resolution is estimated to be about 400 km. Further information about the instrument can be found in Burrows et al. (1995); Bovensmann et al. (1999); Gottwald and Bovensmann (2011).

The retrievals performed for this study were done using version 8 of SCIAMACHY Level 1 data with calibration flags 1, 2, 4, 5 and 7, responsible for the leakage current, pixel-to-pixel gain, straylight correction, wavelength calibration and radiometric calibration, respectively. Level 1 data was provided by the European Space Agency (ESA).

### 2.2 Aerosol parametrization

A balance of the processes creating and removing the stratospheric aerosols as well as of those influencing the size of aerosol

particles commonly results in a size distribution which is well described by a log-normal shape (Hansen and Travis, 1974; Thomason et al., 1997; Deshler et al., 2003; Deshler, 2008; Rieger et al., 2014). Although Deshler et al. (2003) use in some cases the bimodal particle size distribution to achieve the best approximation of their measurements from in-situ optical particle counters (OPC), the same approach is not suitable for remote sensing instruments working in the limb or occultation geometry. This is because 5 independent pieces of information at each altitude level are needed to describe a bimodal log-normal particle

size distribution, while measurements from space borne instruments commonly provide 2 - 3 degrees of freedom per altitude level. Thus, here, similarly to other studies using remote sensing instruments (Thomason et al., 1997; Rieger et al., 2014), the





distribution is considered to be unimodal and is described by

$$n(r) = \frac{N}{\sqrt{2\pi}\ln(\sigma)r} \exp\left(\frac{(\ln(r_g) - \ln(r))^2}{2\ln^2(\sigma)}\right),\tag{1}$$

where $N$ is the total number density of the aerosol particles, $r_g$ is the median radius and $\sigma$ is the standard deviation of the $\ln(n(r))$ function. Though $r_g$ is directly used in the formula and indicated the peak of the distribution in the logarithmic space,

we prefer to work with the mode radius $R_{mod} = r_g/\ln^2(\sigma)$, which determines the peak of the log-normal distribution in the linear space and, thus, is more evident for the interpretation of results. Furthermore, throughout this study a representation of the distribution width by the standard deviation of the size distribution will be used:

$$w = \sqrt{r_g^2 * \exp(\ln^2(\sigma)) * (\exp(\ln^2(\sigma)) - 1)}.\tag{2}$$

Equation (2) represents the absolute distribution width in the linear space, which makes it more convenient value than $\sigma$, as the

latter is relative to the mode radius parameter. In the following text $\sigma$ is used when describing retrieval settings, while absolute distribution width is used when discussing the retrieval results.

## 2.3    Algorithm description

For the retrieval of the aerosol particle size distribution parameters spectral information at 7 wavelengths is used. The latter are chosen outside the intervals near the spectral channel boundaries to avoid artifacts of the optical system. In order to

reduce the measurement noise the sun-normalized radiances at selected wavelengths are averaged in the following intervals $\lambda_1$=750±2 nm, $\lambda_2$=807±2 nm, $\lambda_3$=870±2 nm, $\lambda_4$=1090±2 nm, $\lambda_5$=1235±20 nm, $\lambda_6$=1300±6 nm, $\lambda_7$=1530±30 nm. The wavelengths below 750 nm are not considered because of a lower sensitivity to aerosols, and those above 1530 nm are rejected because of insufficient signal-to-noise ratio. The intervals are chosen to avoid absorption signatures of the atmospheric constituents, thus the uncertainties caused by incorrect absorber profiles knowledge do not affect the retrieval process. Limb

radiances are normalized to the measured solar irradiance spectrum ($\boldsymbol{I_{sol}}$).

Both forward modeling and the retrieval are performed with the software package SCIATRAN-3.8 (Rozanov et al., 2014). Retrieval approach employs an iterative regularized inversion technique similar to that described by Rodgers (2000) to solve the inverse problem. In this approach the inversion of the linearized radiative transfer equation is required, which is formulated as:

$$\boldsymbol{y} - \boldsymbol{y_0} = \mathbf{K}\,\hat{\boldsymbol{x}} + \epsilon,\tag{3}$$

where $\boldsymbol{y} = \ln(\frac{\boldsymbol{I_{mes}}}{\boldsymbol{I_{sol}}})$ is the measurement vector containing the logarithms of the normalized limb radiances at all selected wavelengths and tangent heights, and $\boldsymbol{y_0} = \ln(\frac{\boldsymbol{I_{mod}}}{\boldsymbol{I_{sol}}})$ is a vector, containing the logarithms of the modelled limb spectra. $\mathbf{K}$ stands for the weighting functions or Jacobian matrix. State vector $\hat{\boldsymbol{x}}$ contains relative deviations of the retrieved atmospheric parameters, $\boldsymbol{x}$, from their apriori values, $\boldsymbol{x_0}$, e.g. $[\hat{\boldsymbol{x}}]_i = \frac{[\boldsymbol{x}]_i - [\boldsymbol{x_0}]_i}{[\boldsymbol{x_0}]_i}$ is the $i$-th component of the vector $\hat{\boldsymbol{x}}$. All kinds of errors are

denoted by $\epsilon$.



The solution of the linear inverse problem given by Eq. (3) can be found according to Rodgers (2000) as

$$\hat{\boldsymbol{x}} = (\mathbf{K}^T \mathbf{S_y}^{-1} \mathbf{K} + \mathbf{S_a}^{-1})^{-1} \mathbf{K}^T \mathbf{S_y}^{-1} (\boldsymbol{y} - \boldsymbol{y_0}). \tag{4}$$

Here, $\mathbf{S_a}$ represents the apriori covariance matrix, and $\mathbf{S_y}$ is the noise covariance matrix. Taking into consideration a non-linearity of the inverse problem, an iterative approach is implemented. To decrease the weight of the apriori information, the

apriori state vector $\boldsymbol{x_0}$ is replaced at each iterative step by a state vector obtained at the previous iteration, $\boldsymbol{x_n}$. The state vector variance is constrained to 1% relative to the solution from the previous iteration. Thus the solution at the (n+1) step is given by

$$\tilde{\boldsymbol{x}} = (\mathbf{K_n}^T \mathbf{S_y}^{-1} \mathbf{K_n} + \mathbf{S_a}^{-1})^{-1} \mathbf{K_n}^T \mathbf{S_y}^{-1} (\boldsymbol{y} - \boldsymbol{y_n}), \tag{5}$$

where $[\tilde{\boldsymbol{x}}]_i = \frac{[\boldsymbol{x_{n+1}}]_i - [\boldsymbol{x_n}]_i}{[\boldsymbol{x_n}]_i}$.

The iterative process stops, when the maximum difference between the components of the solution vector at two subsequent iterative steps does not exceed 1%, the relative change of the root mean square between two subsequent iterations is less than 0.001 or limit of 100 iterations is reached. Since the strict constrain is put to deviations from the apriori vector, there are typically 20-30 iterations needed for retrieval process to converge.

Noise covariance matrix is assumed to be diagonal, i.e. the errors are spectrally uncorrelated. Signal-to-noise ratio was

estimated for each scan from SCIAMACHY measurements and varies from 65 to 5000 depending on the wavelength and tangent height. For each of the particle size distribution parameters, the diagonal elements of the apriori covariance matrix are set to 0.01, and the non-diagonal elements drop off exponentially with a correlation radius of 3.3 km, while the elements describing the correlation between different parameters are chosen to be 0.

For the present, only the retrieval of the mode radius $R_{mod}$ and $\sigma$ is performed, and from these parameters absolute distri-

bution width is calculated in accordance with Eq. (2). Number density profile, $N$, remains unchanged for the whole retrieval process, which is justified by a low sensitivity of the retrieval to this parameter (see Sec. 3 for deteils). The first guess parameter values are defined as $R_{mod}$=0.11 $\mu$m and $\sigma$=1.37. The number density profile is selected in accordance with ECSTRA model climatology for the background aerosol loading conditions (Fussen and Bingen, 1999). Scattering phase functions as well as extinction and scattering coefficients per particle are calculated using the Mie scattering theory. Aerosol parameters

are defined from 12 to 46 km, where the particles are assumed to be sulfate droplets (75% $H_2SO_4$ and 25% $H_2O$) with 0% relative humidity in the surrounding atmosphere. Below 12 km and above 46 km the aerosol number density is set to zero. The refractive indices are used from the OPAC database (Hess et al., 1998).

The retrieval is performed for the tropical zone (from 20°S to 20°N) in the altitude range from about 18 km up to about 35 km (the actual altitudes depend on the latitude and season). We focus our initial study at tropics because the transport

mechanisms here are less complicated which makes the interpretation of the obtained results more straightforward. To extent the retrieval to the extra-tropical latitude bands, issues related to differences in observation and illumination conditions need to be dealt with. The choice of the altitude range is justified by lower sensitivity below 18 km (Rieger et al., 2017) and higher biases due to stray light above 35 km. To reduce profile oscillations, the measurement grid is used for the retrieval. The exact





levels of the measurement grid depend on time and latitude, but usually the grid ranges from about 0 km to 100 km with 3.3 km
step. Outside the retrieval range, for the altitudes lower than the minimum retrieval height, the apriori profile scaled relative to
the result at the lowermost retrieved altitude is used, while the altitudes above the upper border are included in the retrieval and
no additional constrains are set.

To reduce the sensitivity of the retrieval to the reflection properties of the underlying surface, effective spectral Lambertian
albedo is concurrently retrieved based on the limb radiances at the same tangent heights, where aerosol particle size retrieval
is performed. As for the particle size parameters, albedo for different wavelengths is uncorrelated with the other parameters
and with the albedo at the other wavelengths. Clouds below and within the field of view remain an issue, even with the albedo
retrieval. Because of that reason just completely cloud free profiles (from 0 km) were used so far. For this research instead
of the standard SCODA cloud filtering algorithm (Eichmann et al., 2016), cloud filtering algorithm based on the statistical
approach was used (Liebing, 2016). The algorithm designed by Liebing (2016) is more preferable to use for aerosol retrieval,
because Eichmann et al. (2016) approach has a disadvantage of tagging the pixels with high aerosol loading (i.e. after the
volcanic eruptions) as cloudy.

Atmospheric pressure and temperature background profiles are from ECMWF (European Center for Medium-Range Weather
Forecasts) operational analysis data for the specific date, time and location of each SCIAMACHY limb measurement.

## 3   Sensitivity studies

### 3.1   Model simulations

In this section sensitivity of the limb radiances to the aerosol particle size distribution parameters is analyzed by performing
simulations with the radiative transfer model SCIATRAN. For this study the model was run for an observational geometry
typical for the tropical region in the middle of July (a solar zenith angle of 41° and solar azimuth angle of 141°). The extrater-
restrial solar flux measured by SCIAMACHY for the chosen day was used. In the radiative transfer model, multiple scattering
was taken into account and the albedo was set to 0 (representation of the ocean surface).

To understand the sensitivity of the different particle size distribution parameters, 3 sets of limb radiance simulations with
consistent variations of parameters were performed. The modelled intensities for a set of particle size distribution parameters at
the tangent height of 25 km are presented in Fig. 1. Lines in Fig. 1 represent natural logarithms of the simulated sun-normalized
intensities for different values of the selected parameter (this representation corresponds to the way, how intensities are treated
by the retrieval algorithm), while shaded areas represent the amplitude of the intensity variations resulting from variations of
the other 2 parameters. In this study following simulation sets were performed: the first set employed a fixed $\sigma$=1.37 and the
same number density profile as used in the retrieval, while the mode radius varied from 0.05 $\mu$m to 0.15 $\mu$m (Fig. 1 A). Another
set of conditions (Fig. 1 B) was simulated by changing $\sigma$ from 1.1 to 2.0 with the fixed $R_{mod}$=0.08 $\mu$m and the same number
density as in previous simulation set. This resulted in the variation of the absolute distribution width from 0.008 to 0.13 $\mu$m.
The last set of simulations (Fig. 1 C) was made by scaling the number density profile with factors from 1 to 2, with the fixed
$R_{mod}$=0.08 $\mu$m and $\sigma$=1.37. The parameter set with standard non-scaled particle number density profile with mode radius





$R_{mod}$=0.08 $\mu$m and $\sigma$=1.37 was chosen as a reference, as this set of parameters is considered to be typical for a background (free of volcanic perturbations) atmosphere.

Analyzing Fig. 1 it is evident, that variations of the limb radiance due to variations of the particle number density ($N$) are remarkably smaller than those due to variations of the mode radius or $\sigma$ / width. The response of the limb radiance to the variation of the number density by a factor of 2 is comparable in the magnitude with that to the variation of mode radius by 0.01 ($\approx$ 13% for given case) and variation of $\sigma$ by 0.13 ($\approx$ 10% for given case). This effect is also illustrated by much smaller weighting functions for particle number density, than those for the mode radius or $\sigma$. The weighting functions ($R_{mod}$, $\sigma$ and $N$) for each retrieved altitude are shown in Fig. 2. As the weighting functions (see Sec. 2.3) show the contribution of the variations in considered parameter to the observed signal, much smaller weighting functions of the particle number density (Fig. 2, right panel) mean much smaller contribution due to variations of the parameter into the measured signal. As a consequence, the particle number density can be neglected in the retrieval and considered to be constant.

It is worth to emphasize again, that the number density profile used in the above described retrieval is derived from the ECSTRA climatology for background aerosol loading conditions (see 2.3). Even though we increase the uncertainty for the volcanic active periods, for the background period the errors are rather small because we use the profile which is expected to be close to the reality. The magnitude of the errors due to unknown number density on the retrieved mode radius and distribution width is discussed in the next section.

## 3.2   Characterization of the retrieval

The effect of the unknown number density on the retrieved mode radius and $\sigma$ as well as general retrieval characteristics are discussed in this section.

To characterize sensitivity of retrieval algorithms averaging kernels are commonly used. In general, this characteristic is specific to the measurement setup, algorithm implementation, and retrieval parameters settings. Averaging kernels for the limb scatter space-borne instruments in the relevant altitude region are distinctly peaked at altitudes, where a bulk of information is originating from. Their shape characterizes the vertical sensitivity and resolution of the measurement-retrieval system and provides an information on the contribution of apriori information to the retrieved profiles.

In Fig. 3 averaging kernels for both $R_{mod}$ (left panel) and $\sigma$ (right panel) typical for the implemented retrieval and tropical observation conditions are presented. Since geometry variations in tropics are rather small, only one specific example for typical illumination and background conditions is presented here. At all altitudes in the retrieval range (in this case 18.5-34.9 km), except for the uppermost one, averaging kernels for both parameters have pronounced peaks at the measurement tangent point altitudes indicating that no significant smearing of the results in the vertical domain is occurred and optimal sensitivity of the retrieval is achieved for each tangent altitude. At the uppermost altitude (34.9 km) averaging kernels are clearly broader and the peak is much less distinct in comparison to the other altitudes. This is an evidence of a strong influence of the neighbouring altitude levels and partial loss of sensitivity.

As pointed out above, in the performed retrieval the relative deviations from the solution obtained at the previous iteration are determined. The state vector variance is set to 1% of the parameter values resulting from the previous iteration. For this





reason such widely used characteristics as the averaging kernel peak value, a posteriori covariance and measurement response are meaningful only within one iterative step but not applicable to the full iterative process.

Another widely used way of assessing the retrieval performance is to simulate the limb radiance using perturbed values for the retrieved parameters and then perform retrieval using synthetic data with the same settings as in the nominal retrieval

process. In this study the simulations are done for an observational geometry typical for SCIAMACHY limb measurements in the tropical region. Additionally, Gaussian noise generator was used to generate 100 independent noise sequences enabling us to evaluate errors resulted from the measurement noise. For our study this approach is very important, because it is essential to analyze the uncertainty resulting from fixed particle number density assumption.

To test the retrieval under different conditions, 5 scenarios were used. All scenarios were simulated for one observation

geometry typical for the tropics. A detailed information on the selected scenarios is presented in the Tab. 1. Generally, the scenarios can be divided into 3 types. The first type includes "small", "background" and "volcanic" scenarios. The intensities for these scenarios were modelled using $R_{mod}$ and $\sigma$ as listed in Tab. 1 with unperturbed particle number density profile, thus the retrieval was run for perturbed $R_{mod}$ and $\sigma$ values. Second type is represented by "volcanic (2N)" scenario, which was modeled with the same $R_{mod}$ and $\sigma$ as the "volcanic scenario", but the particle number density profile was multiplied by factor

of 2 between 12 and 23 km. The perturbation in the number density profile was performed only in the lower layers because significant aerosol loading perturbations due to volcanic eruptions during the SCIAMACHY life time were shown to reach maximum altitude of about 23 km (von Savigny et al., 2015). The third type is represented by the "unperturbed" scenario. For this scenario intensities were simulated with $R_{mod}$=0.11 $\mu$m, $\sigma$=1.37 and unperturbed particle number density profile i.e., with the same parameter values as used as the initial guess for the retrieval. All these values with slight adjustments were taken from

(Deshler, 2008) and represent realistic distributions of the stratospheric aerosol. For all the scenarios modelled surface albedo was perturbed by 0.35.

Figure 4 presents the retrieved profiles of $R_{mod}$ (left panel) and their relative errors (right panel) for the above discussed scenarios. For $\sigma$, retrieved profiles and relative errors are plotted in Fig. 5. Solid lines in the left panels of Figs. 4 and 5 refer to the median retrieved profiles for the scenario, and dashed lines represent true modelled conditions. In the right panels of Figs.

4 and 5 solid lines with the same colors as in the right panels show relative median errors of the retrieved profiles with respect to the true value. In both figures, shaded areas show ±1 standard deviation from the median value.

Analysis of Figs. 4 and 5 shows, that the retrieval results using unperturbed profiles (brown lines) of the mode radius ($R_{mod}$) and $\sigma$ as well as using profiles with the perturbed mode radius ($R_{mod}$) and $\sigma$ (cyan, blue and green lines) are very close to the true values. The relative error is within 20% for $R_{mod}$ and within 5% for $\sigma$. Maxima of the absolute ($\epsilon = |retrieved - true|$)

and relative errors for all scenarios and parameters are summarized in Tab. 1. It is worth to mention, that maximum deviation for the scenarios with unperturbed $N$ is about ±0.01 $\mu$m for $R_{mod}$, for $\sigma$ the absolute error differs, but does not exceed 0.07. For the volcanic scenario with a perturbed number density profile ($N$) it is obvious, that wrong number density assumption influences the retrieved profile, although this influence is rather small. The relative error for that case is less than 20% for mode radius ($R_{mod}$) and less than 10% for $\sigma$, maximum absolute errors for $R_{mod}$ and $\sigma$ are respectively 0.04 $\mu$m and 0.1. The





**Table 1.** Selected scenarios and associated maximum of absolute (relative) errors.

| name | true $R_{mod}$ | true $\sigma$ | color [*] | perturbation | max. $\epsilon_{R_{mod}}$ | max. $\epsilon_\sigma$ | max. $\epsilon_w$ |
|---|---|---|---|---|---|---|---|
| small | 0.06 $\mu$m | 1.7 | cian | $R_{mod}, \sigma$ | 0.01 $\mu$m (19%) | 0.07 (5%) | 0.001 $\mu$m (2%) |
| background | 0.08 $\mu$m | 1.6 | blue | $R_{mod}, \sigma$ | 0.01 $\mu$m(9%) | 0.04 (2%) | $5 \cdot 10^4$ $\mu$m(1%) |
| unperturbed | 0.11 $\mu$m | 1.37 | brown | unpert. | 0.002 $\mu$m(2%) | 0.009 (1%) | $3 \cdot 10^4$ $\mu$m(1%) |
| volcanic | 0.20 $\mu$m | 1.2 | green | $R_{mod}, \sigma$ | 0.01 $\mu$m(5%) | 0.04 (3%) | 0.005 $\mu$m(13%) |
| volcanic (2N) | 0.20 $\mu$m | 1.2 | red | $R_{mod}, \sigma, N$ | 0.04 $\mu$m(19%) | 0.1 (8%) | 0.015 $\mu$m(40%) |

[*] Color of the lines in Figs. 4-6.

In the last 3 columns maximum absolute error for the profile is given by the number without brackets, while the maximum relative error is presented in brackets.

retrieval of scenario with unperturbed particle size distribution parameters resulted in the profiles, which differ within 2% from true values. This characterizes the retrieval error resulting from the measurement noise.

Referring to Sec. 2.2 it is important to remind, that in this study we analyze the absolute distribution width rather than $\sigma$. Even though $\sigma$ as a parameter is widely used in the retrievals and in the climate models, it doesn't provide visually interpretable

information about the width of the particle size distribution, because it is defined relative to the mode radius $R_{mod}$. Due to this reason we use $\sigma$ as a retrieval parameter, while the absolute distribution width as defined by Eq. (2) is used in the interpretation and discussion of the retrieved results. For that reason it is important to asses retrieval uncertainties for the latter parameter either.

In the left panel of Fig. 6, profiles of the absolute distribution width (solid lines), derived from the retrieved $R_{mod}$ and $\sigma$, for

each scenario are depicted. As in the Figs. 4-5, the dashed lines depict the true values. In the right panel of Fig. 6 relative errors are plotted by solid lines. Shaded areas in both panels of Fig. 6 show the standard deviation. Looking at Fig. 6, is obvious, that the retrieval errors from the distribution width for small (cyan lines), background (blue lines) and unperturbed (brown lines) scenarios are rather small: absolute errors (Tab. 1) are less than 0.001 $\mu$m, and relative errors are smaller than 2%. In the volcanic case with unperturbed number density (green lines) differences are within 0.005 $\mu$m, which corresponds to 14%. On

the contrary for the volcanic scenario with the perturbed number density (red lines), the retrieved absolute distribution width deviates from the true value by up to 40% (0.015 $\mu$m).

Although the differences for $R_{mod}$ and $\sigma$ are comparably small, it is important to mention, that from all the modelled scenarios absolute distribution width for the volcanic scenarios is generally the smallest, and larger relative errors are often associated with the division by a small true value. To test this hypothesis another volcanic scenario with larger distribution

width ("wide") was simulated ($R_{mod}$=0.20 $\mu$m and $\sigma$=1.27, number density profile ($N$) is perturbed by the factor of 2 in a layer between 12 and 23 km) and run through the synthetic retrieval. The retrieved profiles of $R_{mod}$, $\sigma$ and absolute distribution width are shown in Fig. 7, where solid lines show the retrieved profiles and dashed lines represent true values. In red, standard "narrow" volcanic scenario with perturbed number density is presented, whereas the magenta colors depict the "wide" scenario (see Tab. 2). Relative errors for these scenarios and parameters are presented in Fig. 8. From Figs. 7 and 8 it is clearly seen that





**Table 2.** Selected scenarios and associated maximum of absolute (relative) errors.

| name | true $R_{mod}$ | true $\sigma$ | color [*] | perturbation | max. $\epsilon_{R_{mod}}$ | max. $\epsilon_\sigma$ | max. $\epsilon_w$ |
|---|---|---|---|---|---|---|---|
| narrow | 0.20 $\mu$m | 1.2 | red | $R_{mod}, \sigma, N$ | 0.04 $\mu$m(19%) | 0.1 (8%) | 0.015 $\mu$m(40%) |
| wide | 0.20 $\mu$m | 1.27 | magenta | $R_{mod}, \sigma, N$ | 0.05 $\mu$m(24%) | 0.1 (8%) | 0.012 $\mu$m(23%) |

[*] Color of the lines in Figs. 7-8.

In the last 3 columns maximum absolute error for the profile is given by the number without brackets, while the maximum relative error is presented in brackets.

even though the behaviour of $R_{mod}$ and $\sigma$ did not change significantly and the absolute/relative errors are very similar for all three parameters, the maximum relative error for the absolute distribution width decreased to 23%.

Summarizing Sec. 3 it can be concluded, that for the background conditions the algorithm is capable to retrieve the mode radius and $\sigma$ with relative accuracy of less than 20% (absolute uncertainty of about 0.01 $\mu$m) and less than 5% (absolute uncertainty less than 0.07), respectively. The relative accuracy of the derived absolute distribution width is dependent on the absolute value of the width and is about 2% (0.001 $\mu$m) for the background conditions. As particle number density is fixed in the retrieval, all uncertainties associated with this parameter are interpreted as changes in the mode radius and/or $\sigma$. Considering previous studies on stratospheric aerosol size distribution, including Deshler et al. (2003); Bingen et al. (2004a, b); Deshler (2008), we believe that for the period from 2002 till 2012 a variation in the aerosol particle number density within a factor of 2 is a realistic assumption. Thus, possible retrieval errors after volcanic eruptions occurred during the SCIAMACHY observation period are estimated to be about 20-25% for the mode radius and less than 10% for $\sigma$. For the volcanic scenarios the relative error for the absolute distribution width can reach up to 40% while the absolute error does not exceed 0.015 $\mu$m.

## 4   Results and discussion

In this section first results of the retrieval of aerosol particle size distribution parameters from SCIAMACHY limb observations are presented. As it was mentioned above (see Sec. 2), the retrieval algorithm was applied to data for the tropical region and completely cloud free (from 0 km) scenes, that resulted in 9727 profiles for the whole SCIAMACHY observation period. The cloud filtering algorithm by Liebing (2016) used in this study employs a probability approach, which means that there is still a 10% chance of cloud or it's fraction to influence the profile. Thus to reject possible unreasonable results, post-retrieval filtering criteria were applied as follows: values with $R_{mod}$ <0.03 $\mu$m were rejected, and aerosol extinction at 750 nm calculated using the retrieved particle size distribution parameters was not allowed to exceed 0.0015 km$^{-1}$. A similar approach to filter the aerosol extinction at 750 nm was used to reject cloudy scenes in SCIAMACHY V1.4 aerosol extinction product (Rieger et al., 2017). Non-converged retrievals with 100 iterations (4.6% of the whole amount of retrieved profiles) were also excluded. No additional filtering criteria were implemented, the maximum mode radius in the retrieved data set reached the value $R_{mod}$=0.21 $\mu$m, and $\sigma$ varied from 1.02 to 2.9. All of the values are considered to be realistic within the reported errors.

The monthly zonal (20°N-20°S) mean values of the mode radii and absolute distribution width are presented in Figs. 9 and 10, respectively. Here, some obvious patterns such as the increase of the values after most of the volcanic eruptions (dashed



lines) can be readily identified. In addition there is a pronounced seasonality of the mode radius ($R_{mod}$) and absolute distribution width. As the seasonal cycle of stratospheric aerosols has already been discussed by several authors (Hitchman et al., 1994; Bingen et al., 2004a), we focus our study on the analysis of the anomalies of the particle size distribution parameters. Anomalies or deseasonalized values for mode radius and absolute distribution width, as shown in Figs. 11 and 12, respectively,

were obtained by subtracting from each monthly mean value an average over all corresponding months in the whole observation period (e.g. the average value for all the Januaries of the 10 year period was subtracted from each January monthly mean value in the observation period).

Analyzing the anomalies for the mode radius presented in Fig. 11 it can be noticed, that there is a distinct increase of the mode radius in the lower altitudes after most of the volcanic eruptions, except for the eruptions of Ruang and Reventador in the

late 2002 and Merapi in late 2010. For these tropical eruptions only a slight increase in the mode radii is observed. This can be related to the small amount of $SO_2$ released during these eruptions (see database of the Smithsonian Institution (2017)). Another important feature is a periodical variation of the mode radius in the 28-32 km range, related to the quasi-biennial oscillation (QBO). A similar QBO signature in the aerosol extinction coefficients retrieved from SCIAMACHY limb measurement at altitudes around 30 km was reported by Brinkhoff et al. (2015) and explained by the influence of the secondary meridional

circulation. QBO pattern is also seen in the anomalies of the distribution width (Fig. 12). Volcanic eruptions influence aerosol particle size distribution width in different ways. As for the mode radius there is an increase of absolute distribution width after Tavurvur, Kasatochi, Sarychev and Merapi eruptions, while a slight decrease of the distribution width at some altitudes after Nabro eruption and no change in the width at all altitudes after Manam eruption is observed.

Following the strong tropical eruptions (Manam, Tavurvur, Nabro) there is a well-defined increase of the mode radius shortly

after the eruptions in the 18 - 21 km altitudes, while at higher altitudes (22-26 km) the volcanic perturbation is observed with a specific time lag. This is the so called tape recorder effect, which is associated with the vertical transport of the air masses in the tropical stratosphere. After strong mid-latitude eruptions (Sarychev, Kasatochi) the increase in mode radius is less pronounced as compared to the tropical volcanoes, though there is a definite increase in the absolute distribution width. There are two possible explanations of this effect. First, the initial and longer term growth in particles is a result of the oxidation of

$SO_2$ to $H_2SO_4$ with chemical rates depending on the physical conditions at a given altitude and latitude. Second, during the transport of the air masses from the mid-latitudes to the tropics the aerosol particle size distribution is modified as a result of sedimentation of heavier particles with large radii. Unfortunately the justification for these hypotheses can be provided only after implementing the current algorithm for the extra-tropical latitudes and modelling the eruptions accounting for the aerosol microphysics, which is outside the scope of this study.

For a better understanding of the variations in the particle size distribution after different volcanic eruptions we considered in more details the temporal evolution of the aerosol particle size distributions at 18, 22 and 25 km altitudes after Manam and Tavurvur eruptions. These distributions are presented in Fig. 13. For ease of comparison, the distributions in the Manam series are plotted in the left column and for Tavurvur in the right column, and $N$=1 cm$^{-3}$ in Eq. (1) was used. In each panel of Fig. 13 four distributions corresponding to different time lags before and after the eruptions obtained at similar latitudes ($\pm 2°$) are

presented. There were 4 time slots chosen: before the eruption (green lines), in the first months after the eruption (red lines),





after more than a half a year after the eruption (blue lines) and in the period of over a year after eruption (cyan lines). The dates corresponding to the curves are presented in the legend below the panels.

As mentioned above, Manam eruption was characterized by an increase of the mode radius, but almost no change of the distribution width after the eruption in comparison to the background conditions. This can be seen, looking at the depicted

distributions. At 18 km (upper left panel in Fig. 13) in the first months after the eruption (red line) the distribution shifts to the larger values in comparison to the distribution before the eruption (green line). After around 7 month after the eruption (blue line) distribution shifts to slightly smaller values, but it's peak is still located at distinctively larger radius value than for the background distribution. At the end of March 2006, 14 month after the eruption (cyan line), the atmosphere at this altitude is "relaxed" and returns to a state close to that before the eruption. It is interesting to mention, that the distribution width after the

eruption (red, blue and cyan lines) does not seem to change much in comparison to that of background conditions (green line), but it is obvious, that the distribution before the eruption (green line) has a distinctively heavier "tail" at the large particle side than the one after the eruption. The distributions at 22 km (middle left panel in Fig. 13) before and shortly after the eruption look similar, the perturbation of the particle size distribution shape is observed 7 and 14 months after eruption (blue and cyan lines). These time delays are attributed to the vertical transport velocity. For the same reason, at 25 km altitude (lower left panel

in Fig. 13) distributions before the eruption as well as 2 and 7 months after the eruption are similar to each other, but the one 14 month after the eruption is shifted to the larger radii. For all the distributions there is no noticeable change observed for the distribution width.

The temporal evolution of the aerosol particle size distribution after the Tavurvur eruption showed a different behaviour than that after Manam. At the lowermost retrieved altitude (18 km, upper right panel) aerosol particle size distribution shifts to the

larger mode radii and gets remarkably wider shortly after the eruption (red line). It appears that the the aerosol formed after the Tavurvur eruption superposed with the one after the weaker Soufriere Hills eruption, which occurred several months before the Tavurvur eruption (see database of Smithsonian Institution (2017)). Although Soufriere Hills eruption might have influenced the observed distributions, it is not possible to distinguish between these two eruptions and we will consider them as one event. In about 8 months after the event, distribution shifts to the smaller mode radii and gets narrower, and in over a year (cyan line)

the distribution returns almost to the same shape as before the eruption (green line), similarly to the evolution after the Manam eruption. At 22 km (middle right panel of Fig. 13) the distribution responses as expected with the time lag of around half a year, and at 25 km the changes are most likely dominated by other processes and not significant in general.

Another remarkable event was the eruption of Nabro in the middle of June of 2011. As presented by Figs. 11 and 12 it was characterized by an increased mode radius and a decreased distribution width. The later might be a result of significant volcanic

activity preceding the Nabro eruption. Since this eruption occurred less than a year before the connection to ENVISAT was lost, we can't fully track it.



## 5   Comparison with SAGE II

As mentioned in Sec. 1, SAGE II was one of the most eminent instruments providing aerosol particle size information. Operating from 1984 till 2005 SAGE II has a 3 years overlap with SCIAMACHY, which enables us to use SAGE II as a comparison instrument in the assessment of SCIAMACHY aerosol particle size distribution product.

In this study the current version v7.0 of the SAGE II stratospheric aerosol data was used (Damadeo et al., 2013). This version of the SAGE II product reports among extinction coefficients at multiple wavelengths such parameter as an effective radius. Assuming the log-normal particle size distribution, effective radius can be related to the particle size distribution parameters as follows:

$$r_{eff} = r_g * \exp(2.5 \ln^2(\sigma)). \tag{6}$$

As SAGE II is a solar occultation instrument, it provides 30 profiles (both sunrise and sunset events) per day. In contrast SCIAMACHY is a limb scatter instrument, providing around 1400 measurements per day. However only tropical and completely cloud free profiles were taken into consideration, so the resulting sampling was much sparser. Taking into consideration this sampling, fairly loose collocation criteria of $\pm 5°$ latitude, $\pm 20°$ longitude and $\pm 24$ hours were used for the comparison, resulting in 57 collocations for 3 years of SCIAMACHY and SAGE II overlap period. As SCIAMACHY vertical sampling is

coarser than the SAGE II one, and SCIAMACHY retrievals were performed at coarser altitude grid (3.3 km opposite to 0.5 km by SAGE II), SAGE II data was first smoothed to the SCIAMACHY vertical resolution and afterwards interpolated on the SCIAMACHY vertical grid.

Mean relative difference between the effective radius from SCIAMACHY and SAGE II is presented in Fig. 14. The relative difference is about 30% in the lower altitudes decreasing with the increasing height to less than 20% at 26 km and around 0%

at 30 km. Possible reasons for the observed lopsided differences are still under investigations. As it can be seen in the Fig. 15, where both collocated data sets are plotted versus time, SCIAMACHY (blue dots) and SAGE II (red dots) effective radii tend to show the same features, however the results from SCIAMACHY are systematically lower at 18 and 21.3 km. To evaluate this aspect more detailed, larger amount of collocations needs to be analyzed.

Summarizing the section it can be concluded, that even though in the lower altitudes the differences are slightly higher, they

are around 30%, which can be considered as a good agreement.

## 6   Conclusions

In this study a retrieval algorithm to obtain two parameters of the stratospheric aerosol particle size distribution (mode radius and $\sigma$) from SCIAMACHY limb measurements was presented. In this retrieval the aerosol particle number density is set to a fixed apriori profile and not retrieved. Wavelength dependent surface albedo is included in the retrieval. The algorithm uses the

measurements of the scattered light at 7 wavelengths and a normalization to the extraterrestrial solar irradiance. Investigation of the averaging kernels showed a good sensitivity in the altitude range from 18 to 32 km for both retrieved parameters. Synthetic retrievals demonstrated that using the presented algorithm for SCIAMACHY limb measurements, errors for the





mode radius are typically in the range 10-20% for unperturbed particle number density profile (the reported absolute error is 0.01 $\mu$m). For $\sigma$, the errors are even smaller and lay within 5% ( absolute error value is less than 0.07). For a perturbed particle number density profile, errors increase to 20% for the mode radius and to 10% for $\sigma$. For easier interpretation of the retrieval results, the absolute distribution widths was used instead of $\sigma$. For the latter parameter, the error is insignificant (less than

2%) for background conditions and can reach 40% after volcanic eruptions, it does not exceed however 0.015 $\mu$m in terms of the absolute differences. Implementation of this retrieval algorithm to SCIAMACHY measurements allowed us to generate the first and unique data set of two particle size distribution parameters from the instrument measuring scattered light in the limb viewing geometry. Analyzing the retrieval results, increase of the mode radius in the altitude range from 18 to 25 km after the volcanic eruptions was identified, while absolute distribution width after the volcanic eruptions did not show any

distinct behaviour. Tape recorder effect or delayed response of the parameters to the volcanic eruptions for higher altitudes was observed for both parameters. Variations in 28-32 km altitude range in both mode radius and absolute distribution width due to quasi biannual oscillation (QBO) were identified. The retrieval results were compared to the SAGE II data and showed an agreement within 30% for effective radii in the lower altitudes, getting better with increasing altitude to better than 10% above 25 km.

**7   Data availability**

SCIAMACHY aerosol particle size distribution data is available after registration at http://www.iup.uni-bremen.de/scia-arc/.

*Competing interests.* The authors declare that they have no conflict of interest.

*Acknowledgements.* This work was funded in parts by European Space Agency (ESA) through SQWG and SPIN projects, German Aerospace Center (DLR) through SADOS project, German Federal Ministry of Education and Research (BMBF) trough ROMIC-ROSA project, Uni-

versity and State of Bremen. We thank ECMWF for providing pressure and temperature information. We also thank NASA for SAGE II data which was downloaded from the NASA Langley Research Center EOSDIS Distributed Active Archive Center.



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



**Figure 1.** Logarithms of sun-normalized intensities spectra at a tangent height of 25 km, modelled with different particle size distribution parameters. Black line represents the "standard" background conditions.





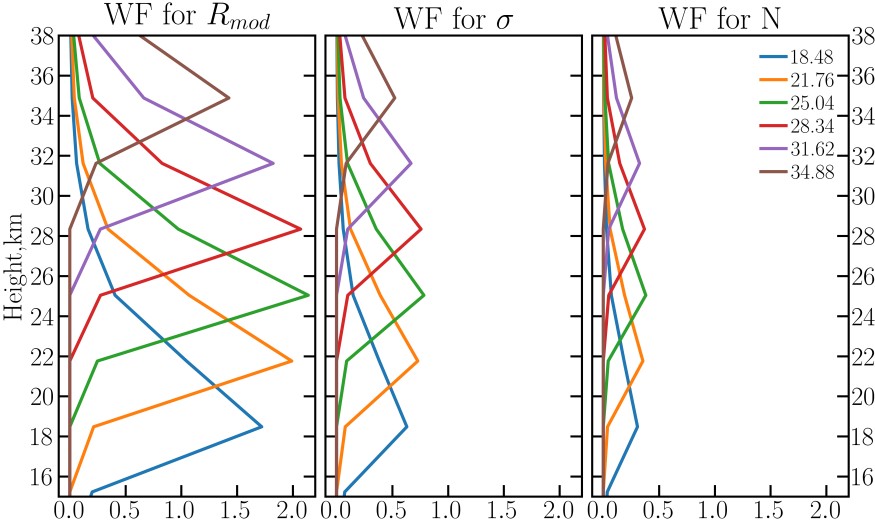

**Figure 2.** Relative logarithmic weighting functions at $\lambda_7$=1530 nm in the retrieval height range for the mode radius $R_{mod}$ (left), $\sigma$ (middle) and particle number density $N$ (right). Colour coded are the different tangent heights.

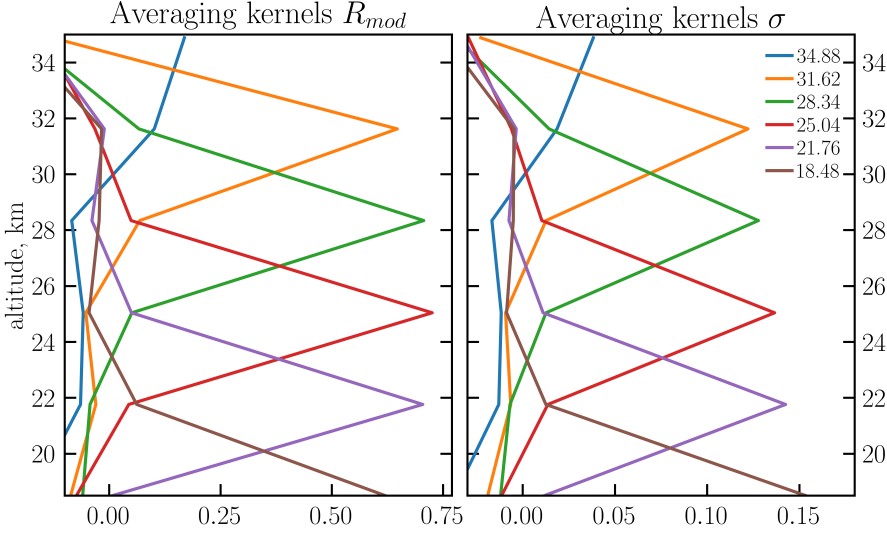

**Figure 3.** Averaging kernels for the aerosol particle size distribution parameters: mode radius $R_{mod}$ (left) and $\sigma$ (right). Results were obtained using the spectra modeled with perturbation of all three parameters. Colour coded are the different tangent heights.



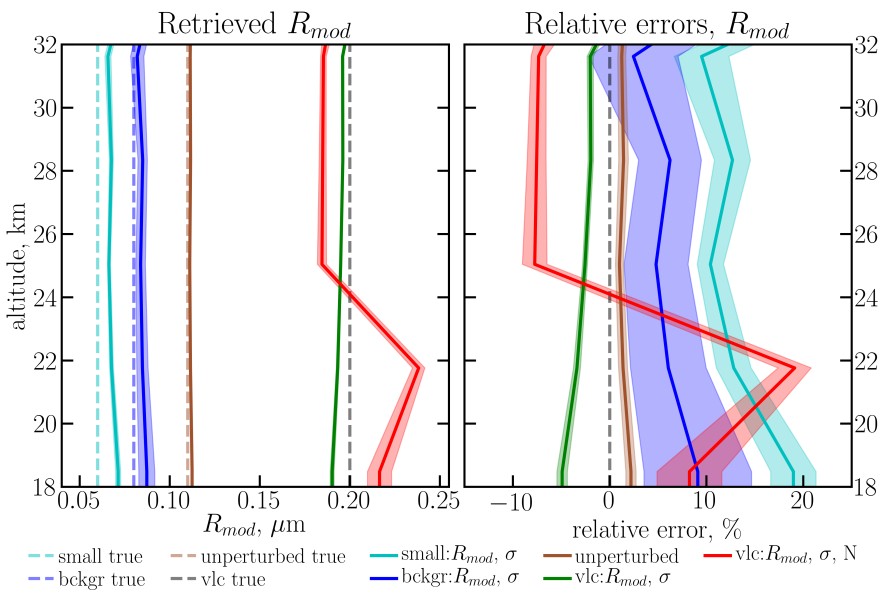

**Figure 4.** Mode radius profiles (solid lines, left panel) and their relative errors (solid lines, right panels) for a typical tropical observation geometry. Synthetic retrievals for unperturbed parameters (brown lines), for small particles (cyan lines) and background case (blue lines) with perturbed $R_{mod}$ and $\sigma$, as well as volcanic case with perturbed mode radius $R_{mod}$ and $\sigma$ (green lines), and with perturbed $R_{mod}$, $\sigma$ and $N$ in a layer (12-23 km) (red lines) with 100 different noise sequences were performed. True values are shown by dashed lines, shaded areas represent standard deviation.





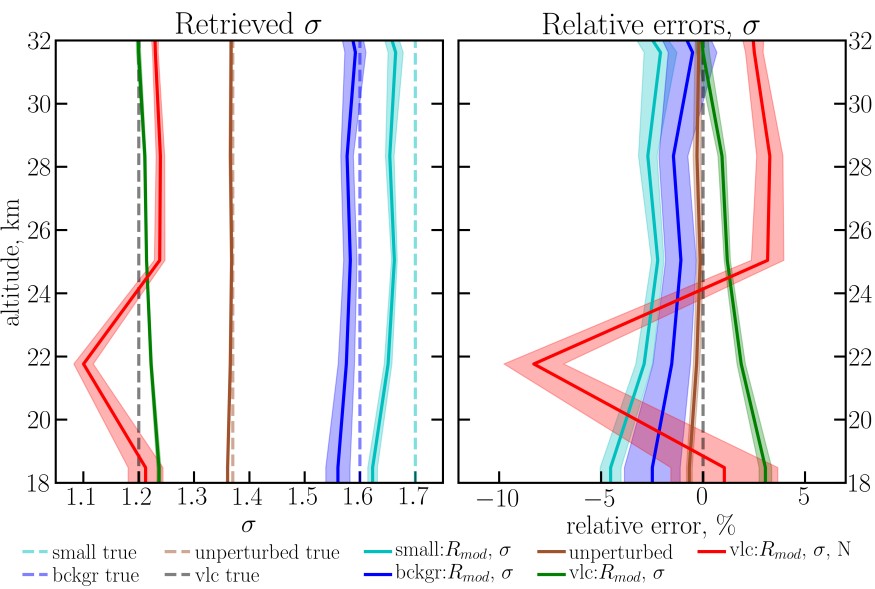

**Figure 5.** $\sigma$ profiles (solid lines, left panel) and their relative errors (solid lines, right panels) for a typical tropical observation geometry. Synthetic retrievals for unperturbed parameters (brown lines), for small particles (cyan lines) and background case (blue lines) with perturbed $R_{mod}$ and $\sigma$, as well as volcanic case with perturbed mode radius $R_{mod}$ and $\sigma$ (green lines), and with perturbed $R_{mod}$, $\sigma$ and $N$ in a layer (12-23 km) (red lines) with 100 different noise sequences were performed. True values are shown by dashed lines, shaded areas represent standard deviation.





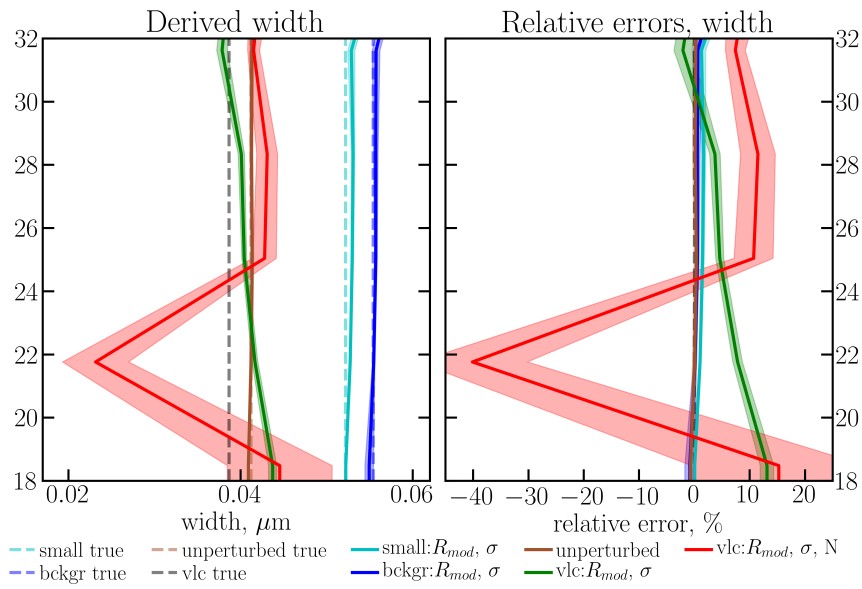

**Figure 6.** Absolute distribution width profiles (solid lines, left panel) and their relative errors (solid lines, right panels) for a typical tropical observation geometry. Synthetic retrievals for unperturbed parameters (brown lines), for small particles (cyan lines) and background case (blue lines) with perturbed $R_{mod}$ and $\sigma$, as well as volcanic case with perturbed mode radius $R_{mod}$ and $\sigma$ (green lines), and with perturbed $R_{mod}$, $\sigma$ and $N$ in a layer (12-23 km) (red lines) with 100 different noise sequences were performed. True values are shown by dashed lines, shaded areas represent standard deviation.




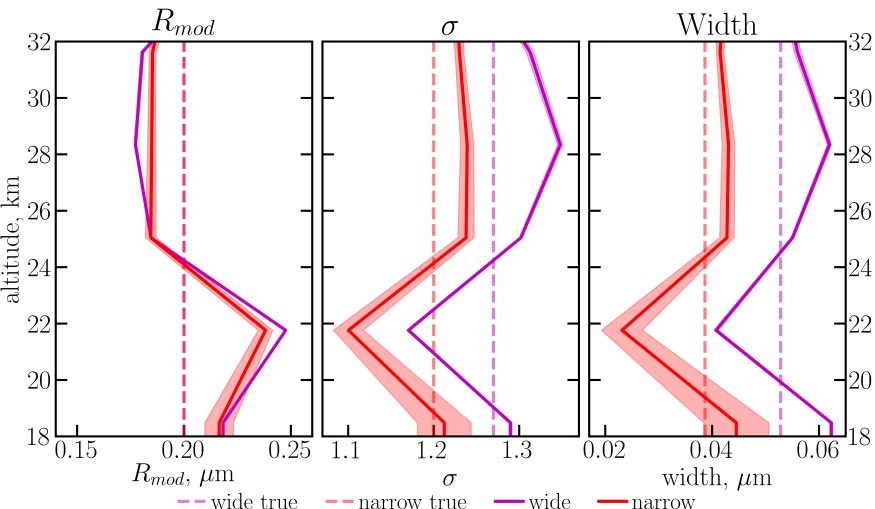

**Figure 7.** Retrieved profiles (solid lines) of $R_{mod}$, $\sigma$ and derived values of the absolute distribution width for the volcanic scenarios with perturbed $R_{mod}$, $\sigma$, $N$. Scenarios with smaller ("narrow", red lines) and larger ("wide", magenta lines) distribution width and 100 different noise sequences were considered. True values are shown by dashed lines, shaded areas represent standard deviation.

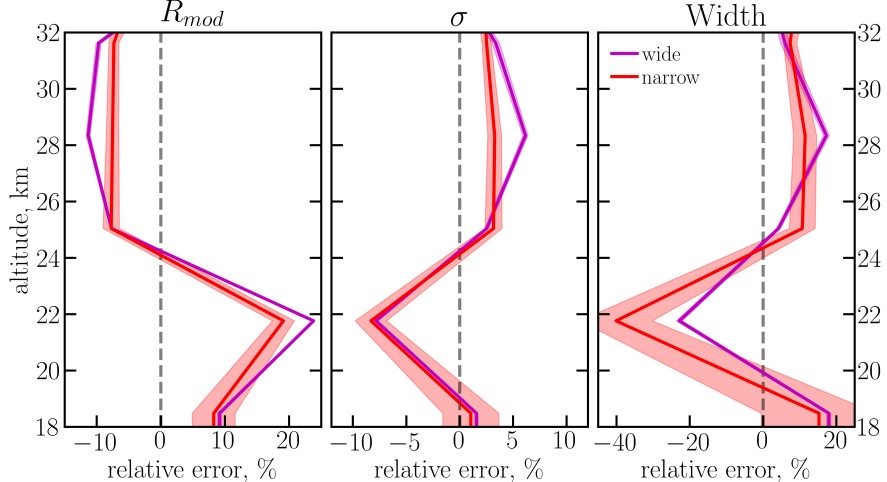

**Figure 8.** Relative errors for $R_{mod}$, $\sigma$ and derived values of the absolute distribution width for the volcanic scenarios with perturbed $R_{mod}$, $\sigma$, $N$. Scenarios with smaller ("narrow", red lines) and larger ("wide", magenta lines) distribution width and 100 different noise sequences were considered. Shaded areas represent standard deviation.





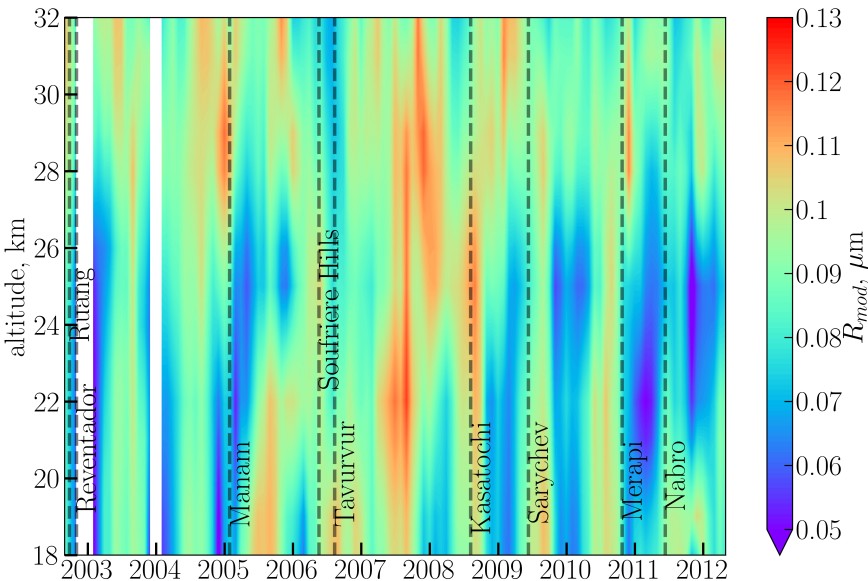

**Figure 9.** Monthly zonal mean values of the stratospheric aerosol mode radius ($R_{mod}$) retrieved from SCIAMACHY limb data in the tropics (20°N - 20°S).

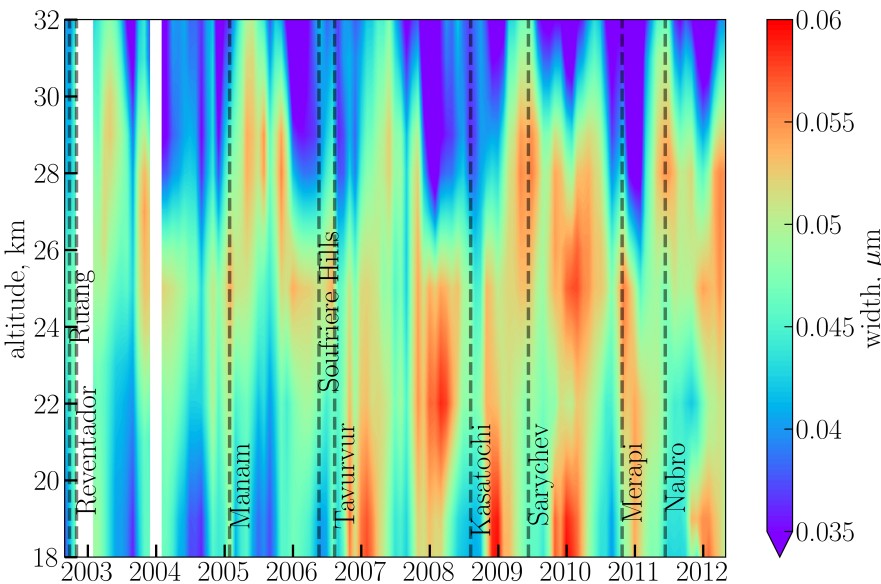

**Figure 10.** Monthly zonal mean values of the stratospheric aerosol absolute distribution width as defined by Eq. (2) retrieved from SCIA-MACHY limb data in the tropics (20°N - 20°S).





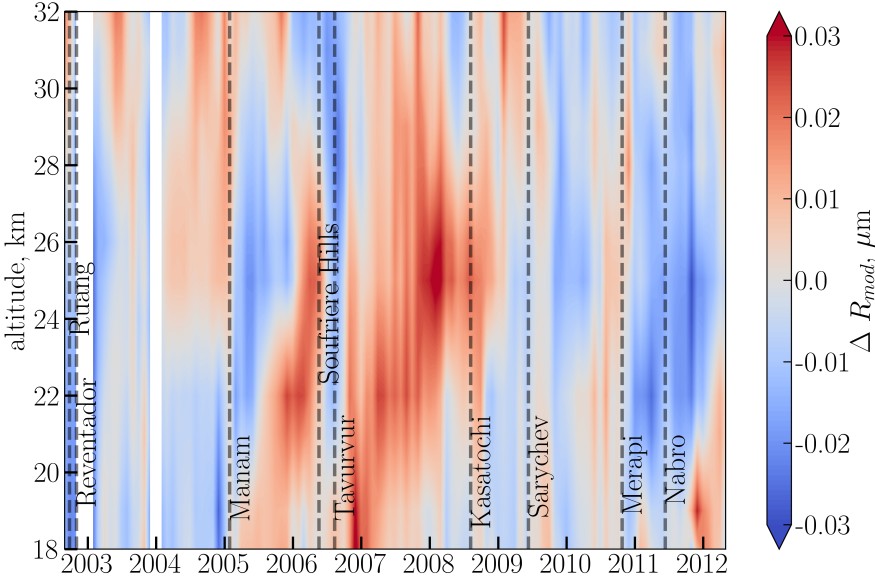

**Figure 11.** Deseasonalized time series (anomalies) of the stratospheric aerosol mode radius ($R_{mod}$) retrieved from SCIAMACHY limb data in the tropics (20°N - 20°S).

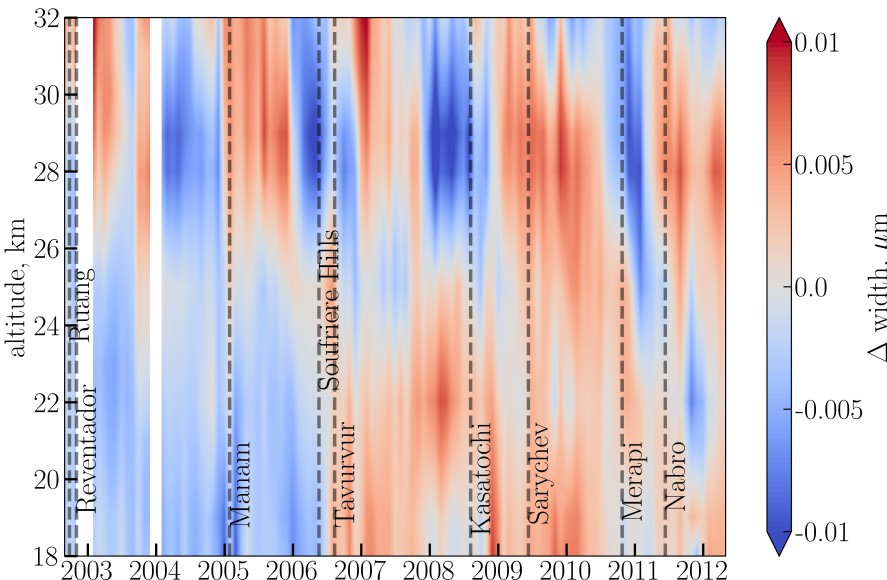

**Figure 12.** Deseasonalized time series (anomalies) of the stratospheric aerosol absolute distribution width as defined by Eq. (2) retrieved from SCIAMACHY limb data in the tropics (20°N - 20°S).





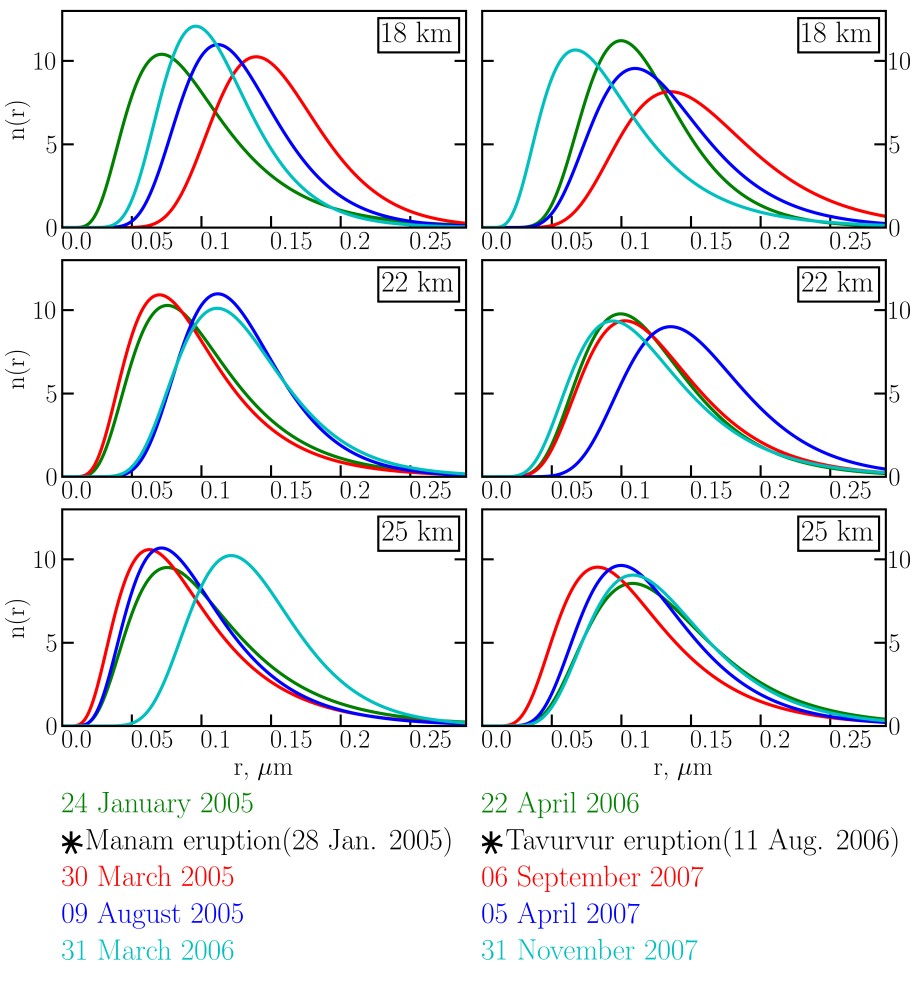

**Figure 13.** Evolution of the aerosol particle size distribution at different altitudes (18, 22, 25 km) after the Manam (left panels) and Tavuvur (right panels) eruptions.



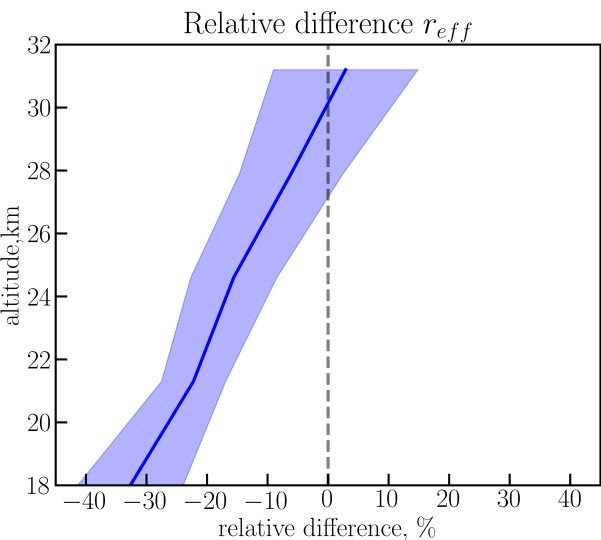

**Figure 14.** Mean relative difference ( (SCIAMACHY-SAGE II) / SAGE II) between effective radii from collocated SCIAMACHY and SAGE II measurements.





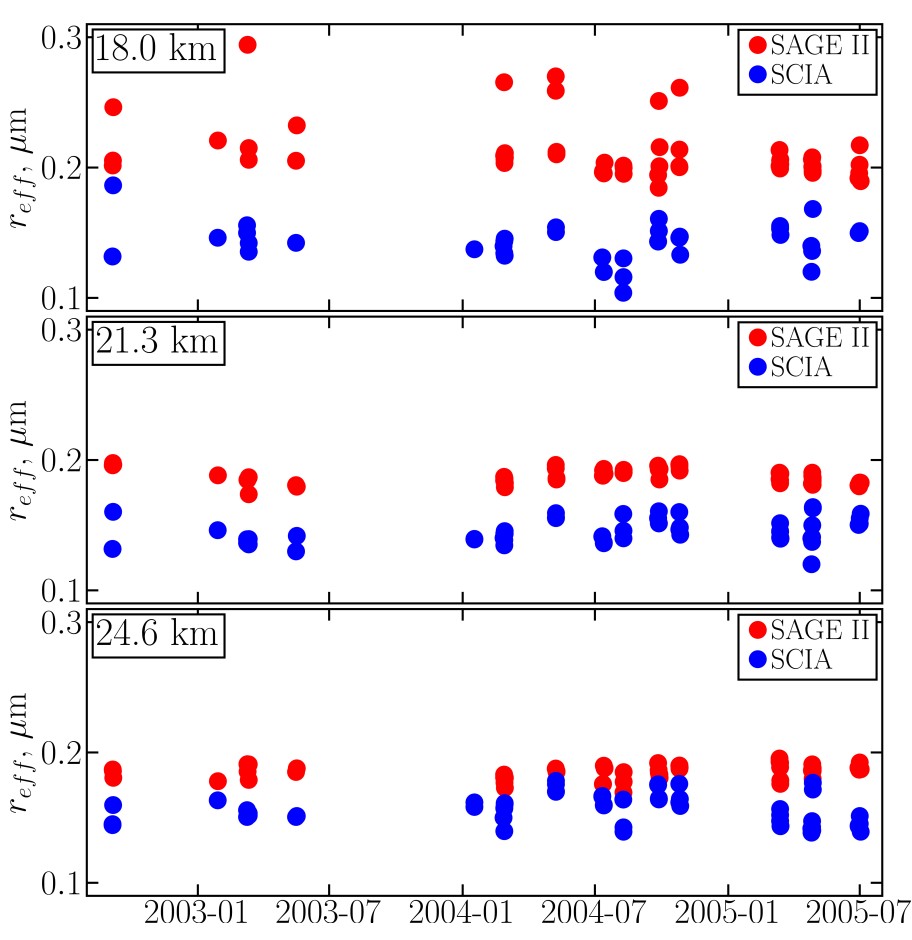

**Figure 15.** Effective radii from collocated SCIAMACHY and SAGE II measurements at 18, 21.3, 24.6 km altitude.