# Peer review of "Aerosol particle size distribution in the stratosphere retrieved from SCIAMACHY limb measurements"

_Atmospheric Measurement Techniques, 2017_

## Referee Comment (RC1) · Anonymous Referee #1 · 8 Dec 2017

**General Comments:**

This paper presents a new algorithm for the retrieval of aerosol particle size distribution (PSD) from SCIAMACHY. As a result of a sensitivity study of the three mode parameters (particle number density N, mode radius Rmod, and standard deviation  $\sigma$ ), the authors conclude that the sensitivity is higher for Rmod and for  $\sigma$  than for N, and in order to alleviate the problem of limited degree of freedom, they decide to retrieve the 2 first parameters and to assign the N parameter using ECSTRA, a climatology published in the literature. This paper is an important milestone since it concerns the first PSD retrieval from limb scattering measurements. The study has been conducted

carefully, but the order of magnitude considered for the various mode parameters may be not well suited, leading, in my opinion, to erroneous conclusions. Fortunately, I don't see any reason why it would invalidate the methodology used. Further details are given below. The authors are invited to revise the English language: sentences are sometimes very long and confusing, and their structures and the use of some word is incorrect (e.g. many confusions between "a" and "the"). It might be useful to let read the manuscript by a native speaker.

Specific comments:

Abstract:

p.1, II. 8-9: It is not true that the aerosol particle number density is unambiguous: it depends on the kind of distribution function used, and on the assumed particle composition. Which approach (PSD or extinction) is optimal depends on the use (e.g. for modelling applications), and it has to be noted that the extinction is basically expressed by the integral of the product of N by the PSD, and since the integration process smooths out all small-scale deviations, the extinction is expected to be a more robust parameter than the PSD mode parameters.

1. Introduction:

p.2, II. 8-9: It is not true that the aerosol particle number density is unambiguous: it depends on the kind of distribution function used, and on the assumed particle composition. Which approach (PSD or extinction) is optimal depends on the use (e.g. for modeling applications), and it has to be noted that the extinction is basically expressed by the integral of the product of N by the PSD, ans since the integration process smoothes out all small-scale deviations, the extinction is expected to be a more robust parameter than the PSD mode parameters.

p.2, I.23: For the sake of completeness, the authors might specify what is the main source of OCS.
p.2, II. 27: Please refer to other sources.

p.2, I. 27: Biomass burning can be of anthropogenic origin. Also Asian anthropogenic sources transported to the stratosphere by the Asian summer monsoon system might be cited.

p.2, II.31-32: I don't understand this "drawback", with respect to PSD retrieval: the PSD retrieval is obviously also determined by the aerosol composition, and the fact that assumptions are made on the PSD in the forward model for limb sounding induces a bias which is obviously a drawback for PSD retrieval.

2. Instrument and applied algorithm:

p.4, II. 24-28: The formulation is misleading. The reasons to use a unimodal lognormal function are a lack of information content and the fact that this function looks reasonable and theoretically acceptable! Ideally, if enough information was available and the aerosol composition was known in detail, a multimodal function taking into account the microphysical properties of all kind of aerosol present would be better.

p.4, l. 29: 6 independent pieces of information are needed.

p.5, l.4, 6, etc.: "linear space" and "logarithmic space" are unclear. Please revise. ("in r", "in log(r)").

p.5, l.13: "spectral information" in unclear: Please specify.

p.5, I.28 : Please define the "weighting functions" of refer to another paper. "Jacobian": of which matrix ?

p.5, I.29: how is the a priori chosen ?

p.6, II. 5-6: There is some contradiction between "state vector obtained at the previous iteration" and "constrained to 1% relative to the solution...". And why the 1% constrain ?

AMTD
p.6, L.17: Please add "[set] arbitrarily", or specify.

p.6, I.25: Above 35 km, it is known that all H2SO4 is only present in gas phase. It is irrelevant to retrieve sulfate aerosol above this altitude.

p.6, I.30-32: Please specify the issues.

p.6, I.33: Please explain.

3. Sensitivity studies:

p.8, II. 3-4, p.9, I.13: Yes it is true for the values chosen here, but another point is to know if these values are chosen in an adequate way. Results published in the literature show that, during the period considered here, N varies between about 10-3 cm-3 at ~20 km and 10-4 cm-3 at ~30 km [Kremser et al., 2016, op. cit.] and even more in other periods [e.g. Deshler et al., 2003, op. cit.]. Taking into account the characteristic dependence of N in altitude (linear in log(N)), this means that N is multiplied by 2 over a distance of about 3 km, which is comparable with SCIAMACHY's resolution. On the other hand, the variation of Rmod and  $\sigma$  is much more gradual in the vertical direction. The choice of the values used in the sensitivity study illustrated in Fig. 1 seems thus not coherent over the 3 parameters. Further, some of the values chosen for Rmod are extremely small (See Bingen et al, 2004, and Deshler et al., 2003, op. cit. for comparison), and I am not fully convinced they are really representative of the typical values found in the lower tropical stratosphere.

p.8, I.11: "Particle number density can be neglected": false ! "and considered to be constant": I don't agree, as explained above. However, I don't see why this would invalidate the methodology used here: it is perfectly acceptable, in my view, to select 2 of the 3 parameters and assign N as done here.

p.8, L. 13: "increase the uncertainty for the volcanic periods": this is certainly not the right way to do: the mean value remains the most probable. I believe this might be a reason, together with the assumptions of background aerosols made for the forward
model, why SCIAMACHY's results show a systematic negative bias with respect to SAGE. (See Ch. 5). p.8, I.26: What do the authors mean by "geometric variations in tropics" ?

p.8, I.31: "This is an evidence...". What is an evidence ?

p.9, l.4: "settings": "characteristics" ?

p.9, I.5: What do the authors mean ? Shouldn't the choice of synthetic data be independent of the instrument configuration ?

p.9-21: Why do the authors distinguish 3 types, while Table 1 gives 5 scenarios ? About "volcanic (2N)" scenario, see rem. Above.

p.9, II.12-34: the word "(un)perturbed" is abundantly used here and this is confusing. Amongst others, the "unperturbed scenario" is perturbed by a noise perturbation. Please use a more clear formulation. In this particular case, "nominal scenario" may help.

p.9, I.21 and figs 4-8: the distributions used here may be realistic at a given location (latitude and altitude), but keeping the vertical profile of PSD constant is not realistic at all ! Please qualify.

p.10, Table 10: The author might usefully add the values of rg and w. If my understanding is right, the values of rg for the 5 scenarios are: {17, 18, 11, 6.6, 6.6}.10-3 micron, what is irrealistically small, and in the size range of condensation nuclei, which seems not realistic to describe the cases mentioned here. As mentioned in Kremser et al., 2016, only median radii > 0.2 micron are observable for instruments like SAGE. SCIAMACHY's spectral range is not very different, and this instrument is not able to observe such small particles. Values of w are found accordingly very small, as {0.12, 0.09, 3.7, 1.24, 1.24}.10-3 micron.

p.10, l.18: If my calculation of w is correct, this is not true.

**AMTD**
p.11, II. 7-12: See above. I don't think that such conclusions can be drawn from the synthetic cases considered here, because I am afraid that they are not representative of the reality.

4. Results and discussion:

p.11, I. 19: Clouds are expected to be composed of very large particles. Thus, why this criteria ?

p.11, I.20: This criteria seems correct in the altitude range where clouds are expected, i.e. up to 2 km above the tropopause. But if fit is applied everywhere, it is likely to exclude important signatures of volcanic plumes (See e.g. Bourassa et al., J. Geophys. Res., 115, 2010).

p.11, II.25- and further: It is very important to realize and to mention that monthly zonal means are not well suited to describe volcanic plumes: such plume fills a very limited part of the whole space and time interval, and the averaging "dilutes" the contribution of the volcanic aerosol load. Hence, it doesn't provide a realistic value of the instantaneous aerosol features, and is always biased (very) low. Averaging is only effective for steady situations.

p.12, l.11: the amount of SO2 was significantly smaller than the other cited eruptions, but was not small. See [Bingen et al., Remote Sensing Env. , 2017] for a recent volcance inventory. Again, the use of monthly means dilutes the effect of the plume and biases the contribution of the eruption low.

p.12, l.30: Are these distributions individual profiles, or monthly zonal means? This shoud be explicitly mentioned, with a reminder of the warning here above.

p.12, I.33: N=1 cm-3: base on what ?

p.13, II. 10-12: What do the authors mean ? For Manam, the values of the green curve at large r are lower than all the other ones ! It is worth to mention that Manam erupted again on 4 March 2006, injecting significant amounts of SO2 up to 17 km height (See

**AMTD**
volcanoe inventory in [Bingen et al., Remote Sensing Env., 2017] ). This new eruption might have influenced the PSD found on 31 March 2006.

- 5. Comparison with SAGE II:
- p.14, II. 18-25: See comment above.
- 6. Conclusions:
- p.15: Conclusions should be revised according to what is discussed above.
- p.15, I.7: The authors should avoid the word "unique", of precise "unique so far".
- p.15, l.9-10: "did not show any distinct behaviour" is meaningless.
- p.15, I.13: Please add a mention on the systematic negative bias.

Technical corrections:

- p.1, l. 7: Please revise the sentence.
- p.1, l.12: "The aerosol particle density is kept constant."
- p.1, l.14: "Overall" instead of "Generally" ?

p.2, l.5: "colossal" : I am not sure this word is used in this context. p.2, l.12: "for modelling the processes"

- p.2, l. 23: Please remove parentheses.
- p.3, l.3: "Otherwise".
- p.3, l.4: 'unambiguously"
- p.3, l.9: "Stratospheric aerosol properties"
- p.3, l.11: Sentence is not correct.
- p.3, II. 19-21: Sentence is not correct.
- p.3, II. 27-29: Please revise the sentence.
- p.3, l.31: "ever" instead of "possible".
- p.4, I.3: "international"; otherwise the country should be mentioned.
- p.4, II. 6: Please revise the sentence.
- p.4, II. 12-16: Please revise the sentences.
- p.4, l.21: "indicating" instead of "responsible" ?
- p. 4, II. 25-26: Please revise the sentence.
- p.4, l. 31: Please revise the sentence.
- p.5, II.9-10: Unclear. Please revise the sentence.
- p.5 and further: "a priori" instead of "apriori" !
- p.6, l.11: "the root mean square between two subsequent iteration": unclear.
- p.7, l.10: "on a statistical approach".
- p.9, l. 7: Please revise the sentence.
- p.10, I.7-8: Please correct the sentence; write "w" instead of "the latter parameter" !
- p.10, l.11: "it is obvious".
- p.13, l.7: "its"
- p.13, I.24: Please revise the sentence.
- p.14, l.15: "compared to" instead of "opposite to".
- p.15, l.7: "an instrument"

---

## Referee Comment (RC2) · Anonymous Referee #2 · 12 Dec 2017

General comments on Malinina et al. (2017):

A lot of good work went into this paper, and it produces several interesting results. But the paper has a few deficiencies that appear to be major, although some may be the result of my misinterpretation. I hope that a combination of clarifying statements and perhaps reassessment of some key statements may resolve these issues.

Specific comments:

Sect. 1, 3rd paragraph:

I have a small concern with the wording of the final sentence: "Biomass burning" is categorized as a "natural" (rather than "anthropogenic") source of stratospheric aerosol precursors. But "biomass burning" is a general term that includes both "natural" and "anthropogenic" fires, so it straddles both categories. This sentence also would benefit from references (both for the proposal that anthropogenic activities contribute significantly to stratospheric aerosol precursors, and for the current consensus that they do not).

Sect. 1, 5th paragraph:

It might be worth mentioning at the end of this paragraph that lidar measurements provide profiles of back-scattering coefficient, which requires further interpretation to determine the extinction coefficient profile.

Sect. 2.1, 2nd paragraph:

Referring to the "horizontal direction" is probably clear enough, but referring to the across-track direction" might be clearer.

Sect. 2.2, 1st paragraph:

A reference for the statement that "space borne instruments commonly provide 2-3 degrees of freedom per altitude level" would be useful. Several space borne instruments exist that use various techniques to observe aerosols, so adding a reference would help to clarify which technique(s) are being described.

Sect. 2.3, 3rd and 4th paragraphs:

The algorithm used in this study is presented as "according to Rodgers (2000)", and equation (4) does correspond to Rodgers (2000) equation (4.5), given the definition of $\hat{x}$ as the deviation from the a priori value. Rodgers (2000) equation (4.5) corresponds to a linear problem, for which no iteration is necessary, but this paragraph describes the non-linearity of the present problem, which requires iteration.

The form that this iteration takes in the present work is expressed in equation (5),

which replaces the a priori state vector with its most recent update at each step of the iteration. This appears to take the approach described by Rodgers (2000) in Sect. 5.6.2, which is entitled "A popular mistake:" It produces slow convergence to a "non-optimal, exact" solution, rather than faster convergence to the "optimal, maximum a posteriori" solution advocated by that author.

So it appears that the solution method used in this study uses the Rodgers (2000) formalism (notation, error analysis, etc.), but in fact the retrieval might be better described as weighted least squares, with the a priori information playing a negligible role (except as a starting point for the iterations). This may be justified, but the reference to Rodgers (2000) without further qualification is misleading, and the authors should explain and justify their methodology more clearly.

One more concern about this area of the text: In the 3rd paragraph, the method is described as constraining "the state vector variance" to "$1\%$ relative to the solution from the previous iteration." But the 4th paragraph states that the iterations stop when "the maximum difference between the components of the solution vector does not exceed $1\%$" (among other criteria). This is confusing: Does the first statement mean that the maximum change in the state vector is $1\%$ for each iteration? If so, then the second statement implies that the iterations would immediately stop, so perhaps it means that the average change cannot exceed $1\%$ for each iteration? This should be clarified.

Sect. 2.3, 6th paragraph:

The text states that "The first guess parameter values are defined as $R_{mod} = 0.11\mu m$ and $\sigma = 1.37$." This follows the description given by von Savigny et al. (2015), Sect. 2.2, which selects a "mono-modal log-normal aerosol particle size distribution with a median radius of 0.11 um and a distribution with of $\sigma = 1.37$ following the in situ balloon observations by Deshler (2008)". And Deshler (2008) shows a bi-modal log-normal size distribution for which the fine mode is characterized by $r_1 = 0.11\mu m$ and $\sigma_1 = 1.37$ in Fig. 3c (as well as a relatively small coarse mode, described by $r_2 = 0.38\mu m$ and

$\sigma_2 = 1.07$).

The parameters $r_1$ and $r_2$ are described as median radii in Deshler (2008), and are used to define the size distribution mathematically in earlier references (e.g., equation (2) of Deshler et al., 2003). This usage defines $r_1$ as equivalent to the parameter $r_g$ defined in equation (1) of the present work, rather than $R_{mod}$ (as defined in the text below that equation).

It therefore seems that either:

a) The text in this paragraph is incorrect, and 0.11 um is used as $r_g$ in equation (1) rather than $R_{mod}$, or

b) The aerosol radius parameter defined by von Savigny et al. (based on the earlier work of Deshler et al.) is being misused in the current work.

This should be clarified in the text, since its implications extend to the entire set of cases described later in Table 1.

Sect. 2.3, 6th and 7th paragraphs:

The 6th paragraph states that "below 12 km and above 46 km the aerosol number density is set to zero." But the 7th paragraph describes scaling that is done to estimate the aerosol outside the retrieval range. I take this to mean that a "scaled" aerosol profile is used from 12 km to the lowest retrieved altitude, and from the highest retrieved altitude to 46 km, and zero aerosol is assumed outside those ranges – is that correct?

Sect. 2.3, 7th paragraph:

This analysis is confined to the tropics. I don't object to this limitation, but the explanation given is that "issues related to differences in observation and illumination conditions need to be dealt with" before other bands can be considered. Some more details about these issues would be useful.

Sect. 2.3, 8th paragraph:

This paragraph states that the "effective spectral Lambertian albedo is concurrently retrieved based on the limb radiances at the same tangent heights where aerosol particle size retrieval is performed." More should be said about how this is done. A frequently used technique (e.g., Sect. V-G of Rault and Loughman, 2013) infers the effective albedo from radiances at tangent heights above the aerosol layer, but that is not consistent with this text.

Sect. 3.1, 3rd paragraph:

The low sensitivity to particle number density ($N$) is a surprising result. As I picture it, the limb radiance for optically thin paths is roughly proportional to the aerosol scattering coefficient in the tangent layer, which is itself proportional to $N$ (for a given size distribution). But this text (as illustrated in Fig. 1) reports that the limb radiance changes by $\approx 10\%$ when $N$ is doubled, throughout the aerosol layer, at 1530 nm (where the line of sight extinction should be dominated by aerosol scattering, due to weak molecular scattering and the relative lack of gaseous absorption).

The relative insensitivity of the aerosol size distribution properties to $N$ plays a key role in the method proposed, so I must be misunderstanding something important about this question. I hope the authors can add some text to explain this point further.

Sect. 3.2, 6th paragraph:

This paragraph ends by stating that "For all the scenarios modelled surface albedo was perturbed by 0.35." Does this mean that the initially assumed surface albedo differs from the true albedo by 0.35 in all cases?

Sect. 3.2, 8th paragraph:

This paragraph describes the relative error in retrieved properties in the sensitivity study. Later (in the last paragraph of this section), some analysis is given describing the expected variability of $N$, based on earlier studies. Has any similar analysis been done to estimate the expected variability of $R_{mod}$ and $\sigma$? That information would

help put the retrieval errors into context better, but of course the lack of a long-term, global data record of these parameters is part of the motivation for this work!

Sect. 4, 6th paragraph:

This paragraph notes changes in the particle distribution function shapes (a heavier "tail", for example) as the Manam aerosol is dispersed. Are all of these distributions single-mode log-normal? And if so, how can these apparent variations in shape occur except through variations of $\sigma$ with time? The last sentence of this paragraph says that $\sigma$ does not change significantly.

Sect. 5, 2nd paragraph:

The Damadeo et al. (2013) reference that describes the V7 SAGE II data refers back to Thomason et al. (2008) to describe their method of computing effective radius. That latter reference mentions log-normal size distributions, but it isn't totally clear that they assume single-mode log-normal distribution when computing the effective radius. This may be relevant as one compares the V7 SAGE II effective radius database to the values derived in this study.

Thomason et al. (2008) also mentions that the SAGE measurements lack sensitivity to aerosols with radius < 100 nm (0.1 um). This might also affect the results of the comparison presented in this section, since the median radius is near 0.1 um for many of the size distributions considered.

Sect. 5, 5th paragraph:

The final sentence summarizes the results of this comparison by calling $30\%$ differences "a good agreement." What standard was used to assess how good this level of agreement is? Does it flow clearly from the estimated error in the SAGE effective radius, in combination with the stated precision of the SCIAMACHY aerosol property error estimates?

Sect. 6:

The effective radius comparison is summarized in the last sentence, noting that agreement improves from $\approx 30\%$ at the bottom of the stratospheric aerosol layer to better than $10\%$ near its top. Might this be related to the fact that the aerosol distribution (as derived by Deshler et al.) is frequently bi-modal at lower altitudes, before becoming a single-mode distribution (as assumed in this study) at higher altitudes?

This leads to a more general question: Were any studies done to define how this algorithm might perform if the true aerosol size distribution is not single-mode lognormal, but instead has some other (plausible) shape? Or if the true aerosol size distribution varies with altitude?

References:

Damadeo, R., Zawodny, J., Thomason, L., and Iyer, N.: SAGE version 7.0 algorithm: application to SAGE II, Atmospheric Measurement Techniques, 6, 3539–3561, 2013.

Deshler, T.: A review of global stratospheric aerosol: Measurements, importance, life cycle, and local stratospheric aerosol, Atmospheric Research, 90, 223–232, 2008.

Deshler, T., Hervig, M., Hofmann, D., Rosen, J., and Liley, J.: Thirty years of in situ stratospheric aerosol size distribution measurements from Laramie, Wyoming (41 N), using balloon-borne instruments, Journal of Geophysical Research: Atmospheres, 108, 2003.

Rault, D. and R.P. Loughman, The OMPS Limb Profiler Environmental Data Record Algorithm Theoretical Basis Document and Expected Performance, IEEE Transactions on Geoscience and Remote Sensing, 51, doi:10.1109/TGRS.2012.2213093, 2013.

Rodgers, C. D.: Inverse methods for atmospheric sounding: theory and practice, vol. 2, World scientific, 2000.

Thomason, L. W., Burton, S. P., Luo, B.-P., and Peter, T.: SAGE II measurements of stratospheric aerosol properties at non-volcanic levels, Atmos. Chem. Phys., 8, 983-995, https://doi.org/10.5194/acp-8-983-2008, 2008.

von Savigny, C., Ernst, F., Rozanov, A., Hommel, R., Eichmann, K.-U., Rozanov, V., Burrows, J., and Thomason, L.: Improved stratospheric aerosol extinction profiles from SCIAMACHY: validation and sample results, Atmospheric Measurement Techniques, 8, 5223–5235, 2015.

---

## Author Comment (AC1) · 7 Feb 2018

We thank the reviewers for the time they spent thoroughly reading the manuscript and constructively commenting on the paper. We answered the reviewers' questions and gave the explanations, which were needed. To distinguish the referees' comments from the author's responses, the comments are shown in italicized font and the responses are highlighted in blue.

*General Comments:*
*This paper presents a new algorithm for the retrieval of aerosol particle size distribution (PSD) from SCIAMACHY. As a result of a sensitivity study of the three mode*

[Figure]

*parameters (particle number density N, mode radius Rmod, and standard deviation $\sigma$), the authors conclude that the sensitivity is higher for Rmod and for $\sigma$ than for N, and in order to alleviate the problem of limited degree of freedom, they decide to retrieve the 2 first parameters and to assign the N parameter using ECSTRA, a climatology published in the literature. This paper is an important milestone since it concerns the first PSD retrieval from limb scattering measurements. The study has been conducted carefully, but the order of magnitude considered for the various mode parameters may be not well suited, leading, in my opinion, to erroneous conclusions. Fortunately, I don't see any reason why it would invalidate the methodology used. Further details are given below. The authors are invited to revise the English language: sentences are sometimes very long and confusing, and their structures and the use of some word is incorrect (e.g. many confusions between "a" and "the"). It might be useful to let read the manuscript by a native speaker.*

Working on the comments of the reviewer, we found a typo in the formula, which defines $R_{mod}$. For that reason the main comment on the unrealistic order of magnitude of $R_{mod}$ and $\sigma$ is resolved, as the reviewer's calculations were done with the wrong formula. Based on the comments we have also realized, that some of our formulations might have been misleading, which resulted in the reviewer's partial misinterpretation of the manuscript. We hope, that the revised manuscript solves these issues. English language was improved in the revised version.

*Specific comments:*
*Abstract:*
*p.1, ll. 8-9: It is not true that the aerosol particle number density is unambiguous: it depends on the kind of distribution function used, and on the assumed particle composition. Which approach (PSD or extinction) is optimal depends on the use (e.g. for modelling applications), and it has to be noted that the extinction is basically*

*expressed by the integral of the product of N by the PSD, and since the integration process smooths out all small-scale deviations, the extinction is expected to be a more robust parameter than the PSD mode parameters.*

Here we meant, that the usage of aerosol particle number density together with aerosol particle size distribution parameters is more optimal approach to describe stratospheric aerosol than its extinction coefficient, because extinction coefficient can be calculated with these parameters but not vice versa. The text has been revised to avoid possible confusion.

*1. Introduction:*
*p.2, ll. 8-9: It is not true that the aerosol particle number density is unambiguous: it depends on the kind of distribution function used, and on the assumed particle composition. Which approach (PSD or extinction) is optimal depends on the use (e.g. for modeling applications), and it has to be noted that the extinction is basically expressed by the integral of the product of N by the PSD, ans since the integration process smoothes out all small-scale deviations, the extinction is expected to be a more robust parameter than the PSD mode parameters.*

The comment repeats the previous one, and in the marked part of the manuscript particle number density is not mentioned. For that reason we have considered this comment as a typo.

*p.2, l.23: For the sake of completeness, the authors might specify what is the main source of OCS.*

The main source of OCS has been specified.

*p.2, ll. 27: Please refer to other sources.*

The information about OCS emissions has been added.

*p.2, l. 27: Biomass burning can be of anthropogenic origin. Also Asian anthropogenic sources transported to the stratosphere by the Asian summer monsoon system might be cited.*

The paragraph has been rewritten in accordance with the reviewer's comments.

*p.2, ll.31-32: I don't understand this "drawback", with respect to PSD retrieval: the PSD retrieval is obviously also determined by the aerosol composition, and the fact that assumptions are made on the PSD in the forward model for limb sounding induces a bias which is obviously a drawback for PSD retrieval.*

We meant, that three particle size distribution parameters describe stratospheric aerosols in more optimal way, than aerosol extinction, because aerosol extinction can be recalculated from the PSD parameters. The paragraph has been rewritten to avoid possible confusion.

*2. Instrument and applied algorithm: p.4, ll. 24-28: The formulation is misleading. The reasons to use a unimodal lognormal function are a lack of information content and the fact that this function looks reasonable and theoretically acceptable! Ideally, if enough information was available and the aerosol composition was known in detail, a multimodal function taking into account the microphysical properties of all kind of*

*aerosol present would be better.*

The paragraph has been rewritten to avoid possible confusion.

*p.4, l. 29: 6 independent pieces of information are needed.* Corrected.

*p.5, l.4, 6, etc.: "linear space" and "logarithmic space" are unclear. Please revise. ("in r", "in log(r)").*

The text has been revised to avoid terms "linear space" and "logarithmic space" instead "$\frac{dn}{dr}$ distribution" and "$\frac{d\ln(n)}{d\ln(r)}$ distribution" have been used. Eq. (1) has been changed respectively.

*p.5, l.13: "spectral information" in unclear: Please specify.*

"Spectral information" has been replaced by "limb radiances".

*p.5, l.28 : Please define the "weighting functions" of refer to another paper. "Jacobian": of which matrix ?*

The paragraph has been revised and the definition has been added.

*p.5, l.29: how is the a priori chosen ?*

The word "a priori [values]" has been replaced by "initial [values]". The choice of the

initial value is described in the following paragraphs.

*p.6, ll. 5-6: There is some contradiction between "state vector obtained at the previous iteration" and "constrained to 1% relative to the solution...". And why the 1% constrain?*

We have rephrased this part of the text to make it less confusing. The statement, that value of 1% was selected empirically to achieve a trade-off between the retrieval stability and sensitivity has been added to the text.

*p.6, L.17: Please add "[set] arbitrarily", or specify.*

The issue has been addressed in the previous comment. The word "empirically" has been added.

*p.6, l.25: Above 35 km, it is known that all H2SO4 is only present in gas phase. It is irrelevant to retrieve sulfate aerosol above this altitude.*

Although there are not so many data above 35 km, that is true, that $H_2SO_4$ is strongly decreasing at higher altitudes and is presented just in the gas phase. For that reason we retrieve aerosol particle size distribution parameters from 18 to 35 km. However, it is known that there is some aerosol above 35 km (e.g. meteoritic dust), but in very small concentrations. It is taken into consideration by our aerosol number density profile, which is about 0.5 cm$^{-3}$ at 35 km and is decreasing exponentially with increasing altitude. We define the aerosol profile above 35 km and below 18 km as sulfate aerosol in order to avoid jumps and unreasonable values at the lowermost and the uppermost retrieval altitudes. In Sect. 3.2 we show, that the trustworthy altitude range is 18-32 km.

[Figure]

*p.6, l.30-32: Please specify the issues.*
The issues have been specified in the revised manuscript.

*p.6, l.33: Please explain.*

In the revised manuscript the phrase "To reduce profile oscillations" has been replaced by "To minimize the need for constrains and avoid additional errors related to e.g. altitude interpolation".

*3. Sensitivity studies:*
*p.8, ll. 3-4, p.9, l.13: Yes it is true for the values chosen here, but another point is to know if these values are chosen in an adequate way. Results published in the literature show that, during the period considered here, N varies between about 10-3 cm-3 at ~20 km and 10-4 cm-3 at ~30 km [Kremser et al., 2016, op. cit.] and even more in other periods [e.g. Deshler et al., 2003, op. cit.]. Taking into account the characteristic dependence of N in altitude (linear in log(N)), this means that N is multiplied by 2 over a distance of about 3 km, which is comparable with SCIAMACHY's resolution. On the other hand, the variation of Rmod and $\sigma$ is much more gradual in the vertical direction. The choice of the values used in the sensitivity study illustrated in Fig. 1 seems thus not coherent over the 3 parameters. Further, some of the values chosen for Rmod are extremely small (See Bingen et al, 2004, and Deshler et al., 2003, op. cit. for comparison), and I am not fully convinced they are really representative of the typical values found in the lower tropical stratosphere.*

Based on this and next comments of the reviewer we realized, that there might be a misunderstanding of the general concept of our study. In our method the particle

number density profile is considered to be known, and $R_{mod}$ and $\sigma$ are retrieved. We believe, that based on the text of the original manuscript the reviewer assumed, that one value of $N$ was used for all altitudes, but that is not the case. A background number density profile from ECSTRA model was used. The text of the manuscript, where particle number density has been mentioned, has been revised to resolve this issue. The profile is changing exponentially from 15.2 cm$^{-3}$ at 18 km to 0.5 cm$^{-3}$ at 35 km, thus, the vertical variations of number density, mentioned by the reviewer, were taken into account. The uncertainty due to changes in the particle number density profile with the time remains an issue, but as it was shown in the Fig. 1 of the manuscript and further with the synthetic studies, the sensitivity of the algorithm to $N$ is rather small. Our number density profile is of the same order of magnitude as the one for $r > 0.15$ presented in Fig. 17 of Kremser et al., (2016). Also in Deshler, (2008) in Fig. 5 the results of the campaign in Darvin, Australia (23 November 2005) are presented. The profiles for $r > 0.15$ are very close to the one, which we are using. Since Darvin is located in the tropical zone, and time of the campaign is within the SCIAMACHY operating period, we believe that used particle number density profile is a good approximation. Also important to mention, that we believe that the variation of number density profile within the factor of 2 is quite realistic for SCIAMACHY observation period, as at the time there were no colossal Pinatubo-like eruptions, all the eruptions were times smaller. According to Fig. 1 in Deshler, (2008), where the particle size distributions 1 year, 3 years and 15 years after Pinatubo eruptions are presented, the variations of $N_1$ (fine mode particle number density) are within a factor of 2 for background and Pinatubo period, and the coarse mode number density ($N_2$) is rather small in comparison to $N_1$).

The reviewer also mentions very small $R_{mod}$ and $\sigma$, but as we realized from the comment to p.10, Table 10, there was a mistake in the formula for $R_{mod}$, which lead to that conclusion. The values for $R_{mod}$ and $\sigma$ are realistic and coincide with the ones from Bingen et al, (2004), and Deshler et al., (2003).

Another important remark, Fig. 1 shows the intensity for one tangent height (25 km),

and thus the altitude variations of the particle size distribution parameters do not play an important role there. The values of $R_{mod}$, $\sigma$ and $N$ near the tangent point are most important.

*p.8, l.11: "Particle number density can be neglected": false ! "and considered to be constant": I don't agree, as explained above. However, I don't see why this would invalidate the methodology used here: it is perfectly acceptable, in my view, to select 2 of the 3 parameters and assign N as done here.*

Please see the answer to the previous comment. Text has been corrected to resolve the issue.

*p.8, L. 13: "increase the uncertainty for the volcanic periods": this is certainly not the right way to do: the mean value remains the most probable. I believe this might be a reason, together with the assumptions of background aerosols made for the forward model, why SCIAMACHY's results show a systematic negative bias with respect to SAGE. (See Ch. 5).*

We disagree with the reviewer on this point. First, as was shown in the manuscript, the $R_{mod}$ errors in the small scenarios are around 0.01 $\mu$m (relative error 10-20%). Implementation of the mean profile would noticeably increase uncertainty for the background cases, while for the volcanic cases, where the relative error is about 20-25%, the uncertainty will remain. Second, there are no known assessments of changes in the particle number density profiles during the Envisat operating period, all the known assumptions were based on SAGE II climatology, which included the colossal eruption of Pinatubo, which is not representative for 2002-2012. SCIAMACHY and SAGE II overlap period is considered to be volcanically quiscent,

thus the lower biases cannot be a result of the particle number density underestimation. This comparison is in general quite problematic. First, we emphasized, that the sample is quite small. Second, as the reviewer mentions, SAGE II is insensitive to the smaller particles typical for the background period, thus, SAGE II effective radius is expected to be biased high. For that reason the relative error of 30% even with systematic negative bias considered to be a good result.

*p.8, l.26: What do the authors mean by "geometric variations in tropics" ?*

The phrase has been replaced by "variations in observational geometry (viewing angle and solar zenith angle)".

*p.8, l.31: "This is an evidence...". What is an evidence ?*

Here we meant the shape of the averaging kernels. The sentence has been changed to avoid a possible confusion.

*p.9, l.4: "settings": "charecteristics" ?*

As we mean the set of parameters, used for the retrieval we think that the word "settings" suits better, than any other.

*p.9, l.5: What do the authors mean ? Shouldn't the choice of synthetic data be independent of the instrument configuration ?*

For the synthetic simulations we chose just one combination of the viewing angle and

solar zenith angle, which are typical for SCIAMACHY measurements in the tropics. In contrast to widely used transmission measurements, the single scattering angle and solar zenith angle for most of the limb viewing instruments, and SCIAMACHY in particular, are dependent on the geographical latitude. This has been clarified in the revised manuscript.

*p.9-21: Why do the authors distinguish 3 types, while Table 1 gives 5 scenarios ? About "volcanic (2N)" scenario, see rem. Above.*

We distinguish 3 types of the simulations, based on the perturbations in the particle size parameters. The explanation has been added to the manuscript.

*p.9, ll.12-34: the word "(un)perturbed" is abundantly used here and this is confusing. Amongst others, the "unperturbed scenario" is perturbed by a noise perturbation. Please use a more clear formulation. In this particular case, "nominal scenario" may help.*

The term "unperturbed" is related to the perturbations in the particle size distribution parameters. To reduce the use of the word "(un)perturbed" and "perturbation" the text has been slightly changed.

*p.9, l.21 and figs 4-8: the distributions used here may be realistic at a given location (latitude and altitude), but keeping the vertical profile of PSD constant is not realistic at all ! Please qualify.*

As explained before, the altitude dependency of the particle number density profile is

accounted for, and now it is clearly stated in the manuscript. For the synthetic studies $R_{mod}$ and $\sigma$ were modelled without altitudinal changes, because there is no information on the realistic behavior of this parameters with the height. Furthermore, the errors have a weak dependency on the specific values. $R_{mod}$ and $\sigma$ used in the study have the same order of magnitude, as the ones from (Desheler et al., 2003; Bingen et al., 2004; Deshler, 2008). We revised the sentence in the manuscript.

*p.10, Table 10: The author might usefully add the values of rg and w. If my under-standing is right, the values of rg for the 5 scenarios are: {17, 18, 11, 6.6, 6.6}.10-3 micron, what is irrealistically small, and in the size range of condensation nuclei, which seems not realistic to describe the cases mentioned here. As mentioned in Kremser et al., 2016, only median radii > 0.2 micron are observable for instruments like SAGE. SCIAMACHY's spectral range is not very different, and this instrument is not able to observe such small particles. Values of w are found accordingly very small, as 0.12,0.09, 3.7, 1.24, 1.24.10-3 micron.*

We are very thankful to the reviewer for that comment, as it helped us to identify a typo in the formula for $R_{mod}$. In reality $R_{mod} = r_g/exp(ln^2(\sigma))$, the formula was corrected in Sect. 2.2. Our calculations though were correct, and $r_g$ and $\sigma$ values are realistic and consistent with other studies (e.g. Deshler et al. (2003); Bingen et al. (2004a, b); Deshler (2008)).

The true values for $r_g$ and $w$ were added to Tab. 1 and 2 of the manuscript.
The reviewer is appealing to Kremser et al. (2016), and authors of that paper are citing Thomason et al. (2008). The introduction of Thomason et al.(2008) ends with a phrase "Since visible and near-infrared wavelength aerosol extinction is insensitive to particles with radii smaller than 100 nm, the robustness of SAD (surface area density) estimates based on these measurements is questionable. Therefore, we have also included a study of the limitations of SAD estimates based on SAGE II

aerosol extinction measurements." We agree, that aerosol extinction in the occultation measurements is insensitive to the small particles ($r < 0.1$ $\mu$m), although for the limb scattering measurements the situation is slightly different. As can be found e.g. in Chandrasekhar (1960) for the scattered light $I \sim \sigma_s N p$, while for the transmission $\ln I \sim \sigma_s N$, where $I$ is intensity, $\sigma_s$ is aerosol extinction cross section, and $p$ is the scattering phase function, which depends on $(R_{mod}, \sigma)$. Thus, limb radiances are differently sensitive to the aerosol parameters than the occultation measurements. As an example we simulated the intensities for 3 different distributions: $R_{mod}$=0.06 $\mu$m, $R_{mod}$=0.08 $\mu$m, $R_{mod}$=0.10 $\mu$m as well as the intensities with no aerosol. For all distributions $\sigma$ was chosen so, that $w \approx$0.01 $\mu$m. We calculated the relative differences of the intensities (($\Delta I = I_{aer} - I_{noaer}) * 100/I_{noaer}$). This relative differences are plotted in the Fig. 1 in the supplement to the answer. As it can be seen from the Fig. 1, for the distribution with $R_{mod}$=0.06 $\mu$m $\Delta I$ is 1% which is around the sensitivity threshold, although for the distributions with $R_{mod}$=0.08 $\mu$m, $R_{mod}$=0.10 $\mu$m $\Delta I$ is about 5 and 15% respectively. Thus, we believe, that SCIAMACHY limb measurements are more sensitive to the smaller particles than SAGE II.

*p.10, l.18: If my calculation of w is correct, this is not true.*

The calculations of the reviewer were not correct, as there was a typo in the formula (see previous comment).

*p.11, ll. 7-12: See above. I don't think that such conclusions can be drawn from the synthetic cases considered here, because I am afraid that they are not representative of the reality.*

As mentioned in the two previous comments, there was a typo in the formula, and

because of that the calculations of the reviewer were not correct. The scenarios are realistic and are consistent with Deshler et al. (2003); Bingen et al. (2004a, b); Deshler (2008).

*4. Results and discussion:*
*p.11, l. 19: Clouds are expected to be composed of very large particles. Thus, why this criteria ?*

This criterion was chosen to eliminate the distributions with the unrealistic values rather than clouds. As it was mentioned in the comment to "p.10, Table 10", the distributions with $R_{mod}$=0.06 $\mu$m and $\sigma \approx$1.1 is on the lower limit of the sensitivity of SCIAMACHY measurements, but in reality this distribution is highly unprobable. According to our data for the distributions with $R_{mod}$=0.06 $\mu$m mean $\sigma$ value is around 1.7. For the distributions with $R_{mod}$=0.03 $\mu$m and $\sigma$ around 1.6 the threshold $\Delta I$ value of 1% is reached. We have changed the text of the manuscript to make this paragraph more clear.

*p.11, l.20: This criteria seems correct in the altitude range where clouds are expected, i.e. up to 2 km above the tropopause. But if fit is applied everywhere, it is likely to exclude important signatures of volcanic plumes (See e.g. Bourassa et al., J. Geophys. Res., 115, 2010).*

The criterion was applied everywhere, however, as follows from Fig. 1 in (Bourassa et al., 2010), the daily mean values for aerosol extinction at 750 nm never reached the value higher than 0.0015 even after the eruption. In their research Bourassa et al.(2010) use 0.0015 as the highest value for their color bar, thus we believe that this value will not exclude volcanic plumes.

*p.11, ll.25- and further: It is very important to realize and to mention that monthly zonal means are not well suited to describe volcanic plumes: such plume fills a very limited part of the whole space and time interval, and the averaging "dilutes" the contribution of the volcanic aerosol load. Hence, it doesn't provide a realistic value of the instantaneous aerosol features, and is always biased (very) low. Averaging is only effective for steady situations.*

In this manuscript we do not aim to analyze volcanic plumes, but rather to understand the general state of the atmosphere in 2002-2012, and for that purpose monthly mean values serve quite well. We added a statement about the purpose of the evaluation, and the issues of the monthly mean values in the revised manuscript.

*p.12, l.11: the amount of SO2 was significantly smaller than the other cited eruptions, but was not small. See [Bingen et al., Remote Sensing Env. , 2017] for a recent volcanoe inventory. Again, the use of monthly means dilutes the effect of the plume and biases the contribution of the eruption low.*

The word "small" was changed to "smaller" and citation to Bingen et al.(2017) was added.

*p.12, l.30: Are these distributions individual profiles, or monthly zonal means ? This shoud be explicitely mentioned, with a reminder of the warning here above.*

The distributions are individual profiles. This is explicitly mentioned in the revised manuscript.

*p.12, l.33: N=1 cm-3: base on what ?*

As $N$ is just a multiplicative factor of $n(r)$, and $N$ is decreasing exponentially with the height, we chose $N$=1 cm$^{-3}$ for each altitude to show the form of the distributions with the same y-axis for all altitudes. The explanation has been added to the text.

*p.13, ll. 10-12: What do the authors mean ? For Manam, the values of the green curve at large r are lower than all the other ones ! It is worth to mention that Manam erupted again on 4 March 2006, injecting significant amounts of SO2 up to 17 km height (See volcanoe inventory in [Bingen et al., Remote Sensing Env., 2017] ). This new eruption might have influenced the PSD found on 31 March 2006.*

By "heavier tails" we meant "stronger relative contribution of the larger particles". In the revised manuscript this formulation has been corrected.
In the revised manuscript we also mention the eruption on 4 March 2006. This explains, why at 18 km the cyan line did not return to the background state as indicated by the green line.

*5. Comparison with SAGE II:*
*p.14, ll. 18-25: See comment above.*

Please, see the answer to that comment.

*6. Conclusions:*
*p.15: Conclusions should be revised according to what is discussed above.*

[Figure]

Conclusions have been slightly revised. However as we believe the main issue raised by the reviewer has been clarified, we have not done any major revisions.

*p.15, l.7: The authors should avoid the word "unique", of precise "unique so far".*

Changed to "for now unique".

*p.15, l.9-10: "did not show any distinct behaviour" is meaningless.*

The clarification of this phrase is provided in the revised manuscript.

*p.15, l.13: Please add a mention on the systematic negative bias.*

The systematic negative bias has been mentioned.

*Technical corrections:*
*p.1, l. 7: Please revise the sentence.* The sentence has been revised.

*p.1, l.12: "The aerosol particle density is kept constant."*

The suggested phrase might be misinterpreted as keeping the particle number density constant with the height. This is why we left the original phrase adding the word "profile": "the aerosol particle number density profile remains unchanged".

*p.1, l.14: "Overall" instead of "Generally" ?* Has been replaced by "typically".

*p.2, l.5: "colossal" : I am not sure this word is used in this context.*

According to the database of the Smithsonian Institution, cited in the manuscript, and other sources Mount Pinatubo eruption 1991 had VEI=6, which is classified as colossal eruption.

*p.2, l.12: "for modelling the processes"*Corrected.

*p.2, l. 23: Please remove parentheses.*Corrected.

*p.3, l.3: "Otherwise".* Has been change to "Additionally".

*p.3, l.4: 'unambiguously"*As we haven't found any place, where this word occurs at the marked page and line, we cannot address this comment.

*p.3, l.9: "Stratospheric aerosol properties"*The sentence has been revised.

*p.3, l.11: Sentence is not correct.*Corrected.

*p.3, ll. 19-21: Sentence is not correct.*Corrected

*p.3, ll. 27-29: Please revise the sentence.*Revised.

*p.3, l.31: "ever" instead of "possible".* The sentence has been rephrased, and "possible" has been replaced by "among".

*p.4, l.3: "international"; otherwise the country should be mentioned.* Corrected.

*p.4, ll. 6: Please revise the sentence.* Corrected.

*p.4, ll. 12-16: Please revise the sentences.* Revised.

*p.4, l.21: "indicating" instead of "responsible" ?* Corrected.

*p. 4, ll. 25-26: Please revise the sentence.* Revised.

*p.4, l. 31: Please revise the sentence.* Revised

*p.5, ll.9-10: Unclear. Please revise the sentence.* Revised.

*p.5 and further: "a priori" instead of "apriori" !* Corrected.

*p.6, l.11: "the root mean square between two subsequent iteration": unclear.* Sentence has been revised.

*p.7, l.10: "on a statistical approach".* Corrected.

[Figure]

*p.9, l. 7: Please revise the sentence.* The sentence has been removed.

*p.10, l.7-8: Please correct the sentence; write "w" instead of "the latter parameter" !* Corrected.

*p.10, l.11: "it is obvious".* Corrected.

*p.13, l.7: "its"* Corrected.

*p.13, l.24: Please revise the sentence.* Revised.

*p.14, l.15: "compared to" instead of "opposite to".* Corrected.

*p.15, l.7: "an instrument"* Corrected.
* * *
Done.

[Figure]

**Fig. 1.** The relative differences of the intensities (($\Delta$I = Iaer $-$ Inoaer ) * 100/Inoaer ) for the scenarios with and without aerosol loading. Rmod and $\sigma$ for the scenarios are shown in the legend.

---

## Author Comment (AC2) · 7 Feb 2018

We thank the reviewers for the time they spent thoroughly reading the manuscript and constructively commenting on the paper. We answered the reviewers' questions and gave the explanations, which were needed. To distinguish the referees' comments from the author's responses, the comments are shown in italicized font and the responses are highlighted in blue.

*General comments on Malinina et al. (2017):*
*A lot of good work went into this paper, and it produces several interesting results. But the paper has a few deficiencies that appear to be major, although some may be the*

*result of my misinterpretation. I hope that a combination of clarifying statements and perhaps reassessment of some key statements may resolve these issues.*

*Specific comments:*
*Sect. 1, 3rd paragraph:*
*I have a small concern with the wording of the final sentence: "Biomass burning" is categorized as a "natural" (rather than "anthropogenic") source of stratospheric aerosol precursors. But "biomass burning" is a general term that includes both "natural" and "anthropogenic" fires, so it straddles both categories. This sentence also would benefit from references (both for the proposal that anthropogenic activities contribute significantly to stratospheric aerosol precursors, and for the current consensus that they do not).*

The paragraph and formulations have been revised.

*Sect. 1, 5th paragraph:*
*It might be worth mentioning at the end of this paragraph that lidar measurements provide profiles of back-scattering coefficient, which requires further interpretation to determine the extinction coefficient profile.*

The information about the backscatter coefficient has been added.

*Sect. 2.1, 2nd paragraph:*
*Referring to the "horizontal direction" is probably clear enough, but referring to the across-track direction" might be clearer.*

Wording "horizontal direction" has been changed to "horizontal cross-track direction".

*Sect. 2.2, 1st paragraph:*
*A reference for the statement that "space borne instruments commonly provide 2-3 degrees of freedom per altitude level" would be useful. Several space borne instruments exist that use various techniques to observe aerosols, so adding a reference would help to clarify which technique(s) are being described.*

The text has been modified and the references have been added.

*Sect. 2.3, 3rd and 4th paragraphs:*
*The algorithm used in this study is presented as "according to Rodgers (2000)", and equation (4) does correspond to Rodgers (2000) equation (4.5), given the definition of $\hat{x}$ as the deviation from the a priori value. Rodgers (2000) equation (4.5) corresponds to a linear problem, for which no iteration is necessary, but this paragraph describes the non-linearity of the present problem, which requires iteration.*
*The form that this iteration takes in the present work is expressed in equation (5), which replaces the a priori state vector with its most recent update at each step of the iteration. This appears to take the approach described by Rodgers (2000) in Sect. 5.6.2, which is entitled "A popular mistake:" It produces slow convergence to a "non-optimal, exact" solution, rather than faster convergence to the "optimal, maximum a posteriori" solution advocated by that author.*
*So it appears that the solution method used in this study uses the Rodgers (2000) formalism (notation, error analysis, etc.), but in fact the retrieval might be better described as weighted least squares, with the a priori information playing a negligible role (except as a starting point for the iterations). This may be justified, but the reference to Rodgers (2000) without further qualification is misleading, and the authors should explain and justify their methodology more clearly.*

[Figure]

*One more concern about this area of the text: In the 3rd paragraph, the method is described as constraining "the state vector variance" to " 1% relative to the solution from the previous iteration." But the 4th paragraph states that the iterations stop when "the maximum difference between the components of the solution vector does not exceed 1% " (among other criteria). This is confusing: Does the first statement mean that the maximum change in the state vector is 1% for each iteration? If so, then the second statement implies that the iterations would immediately stop, so perhaps it means that the average change cannot exceed 1% for each iteration? This should be clarified.*

We corrected the paragraph, so the Rodgers' approach is emphasized. We revised the text and stated, that our retrieval is "the weighted regularized inversion similar to 0-order Tikhonov method". We also added a justification of the choice of this method instead of the Rodgers' optimal estimation.
We agree with the reviewer, that the formulations in the paragraph about the changes of the a priori information was misleading. We have improved it, and hope, that in the revised manuscript it is clear, that the solution $x_{n+1}$ can reach any possible value, and the variance contributes to the cost function and is not a hard constrain.

*Sect. 2.3, 6th paragraph:*
*The text states that "The first guess parameter values are defined as $R_{mod} = 0.11 \mu m$ and $\sigma = 1.37$." This follows the description given by von Savigny et al. (2015), Sect.2.2, which selects a "mono-modal log-normal aerosol particle size distribution with a median radius of 0.11 um and a distribution with of $\sigma = 1.37$ following the in situ balloon observations by Deshler (2008)". And Deshler (2008) shows a bi-modal log-normal size distribution for which the fine mode is characterized by $r_1 = 0.11 \mu m$ and $\sigma_1 = 1.37$ in Fig. 3c (as well as a relatively small coarse mode, described by $r_2 = 0.38 \mu m$ and $\sigma_2 = 1.07$).*

*The parameters $r_1$ and $r_2$ are described as median radii in Deshler (2008), and are used to define the size distribution mathematically in earlier references (e.g., equation (2) of Deshler et al., 2003). This usage defines $r_1$ as equivalent to the parameter $r_g$ defined in equation (1) of the present work, rather than $R_{mod}$ (as defined in the text below that equation).*
*It therefore seems that either:*
*a) The text in this paragraph is incorrect, and 0.11 um is used as $r_g$ in equation (1) rather than $R_{mod}$, or*
*b) The aerosol radius parameter defined by von Savigny et al. (based on the earlier work of Deshler et al.) is being misused in the current work.*
*This should be clarified in the text, since its implications extend to the entire set of cases described later in Table 1.*

The first guess parameters were set arbitrarily, so the parameters are $R_{mod} = 0.11 \mu m$ and $\sigma = 1.37$ ($r_g$=0.12). The text have been revised to avoid a possible confusion. Here we do not cite neither von Savigny et al. (2015), nor Deshler (2008), so their works have not been misused. In our algorithm the first guess parameters do not play any significant role, and our internal test have shown, that changes in the first guess parameters (e.g. $R_{mod}$=0.13 $\mu$m or $R_{mod}$=0.15 $\mu$m) influence the retrieved results by less than 1%.

*Sect. 2.3, 6th and 7th paragraphs:*
*The 6th paragraph states that "below 12 km and above 46 km the aerosol number density is set to zero." But the 7th paragraph describes scaling that is done to estimate the aerosol outside the retrieval range. I take this to mean that a "scaled" aerosol profile is used from 12 km to the lowest retrieved altitude, and from the highest retrieved altitude to 46 km, and zero aerosol is assumed outside those ranges – is that correct?*

The reviewer's understanding is correct. The text has been slightly modified to avoid possible confusion.

*Sect. 2.3, 7th paragraph:*
*This analysis is confined to the tropics. I don't object to this limitation, but the explanation given is that "issues related to differences in observation and illumination conditions need to be dealt with" before other bands can be considered. Some more details about these issues would be useful.*

The issues have been specified.

*Sect. 2.3, 8th paragraph:*
*This paragraph states that the "effective spectral Lambertian albedo is concurrently retrieved based on the limb radiances at the same tangent heights where aerosol particle size retrieval is performed." More should be said about how this is done. A frequently used technique (e.g., Sect. V-G of Rault and Loughman, 2013) infers the effective albedo from radiances at tangent heights above the aerosol layer, but that is not consistent with this text.*

As we show it in Fig. 3 and Sect. 3.2 at 35 km aerosol influence at the radiances is rather small, thus the information from this tangent altitude contributes mainly to the albedo retrieval, and other tangent altitudes are employed for the stability reason. This explanation has been added to the text.

*Sect. 3.1, 3rd paragraph:*

*The low sensitivity to particle number density ($N$) is a surprising result. As I picture it, the limb radiance for optically thin paths is roughly proportional to the aerosol scattering coefficient in the tangent layer, which is itself proportional to $N$ (for a given size distribution). But this text (as illustrated in Fig. 1) reports that the limb radiance changes by $\approx 10\%$ when $N$ is doubled, throughout the aerosol layer, at 1530 nm (where the line of sight extinction should be dominated by aerosol scattering, due to weak molecular scattering and the relative lack of gaseous absorption).*
*The relative insensitivity of the aerosol size distribution properties to $N$ plays a key role in the method proposed, so I must be misunderstanding something important about this question. I hope the authors can add some text to explain this point further.*

There was some misunderstanding about the influence of $N$ on the intensities. In the manuscript we meant, that changes in $N$ by 200% have the same influence on the intensities as 13% change in $R_{mod}$ or 10% change in $\sigma$. This place in the manuscript has been revised to avoid possible confusion. As can be found in the text books on radiative transfer ( e.g. Chandrasekhar,1960) for measurements of the scattered light intensity $I \sim \sigma_s N p$, where $p = f(R_{mod}, \sigma)$ is the scattering phase function and $\sigma_s = f(R_{mod}, \sigma)$ is aerosol extinction cross section. The dependency of $p$ on $R_{mod}$ and $\sigma$ is stronger, than linear. For that reason $R_{mod}$ and $\sigma$ have stronger contribution to $I$, than $N$.

*Sect. 3.2, 6th paragraph:*
*This paragraph ends by stating that "For all the scenarios modelled surface albedo was perturbed by 0.35." Does this mean that the initially assumed surface albedo differs from the true albedo by 0.35 in all cases?*

We have reformulated the text of the manuscript and added in the algorithm description the initial guess for albedo (0.5), and in the Sect. 3.2 "true" modelled value (0.15).

We also have added information about the retrieved values of albedo for the synthetic retrievals.

*Sect. 3.2, 8th paragraph:*
*This paragraph describes the relative error in retrieved properties in the sensitivity study. Later (in the last paragraph of this section), some analysis is given describing the expected variability of $N$, based on earlier studies. Has any similar analysis been done to estimate the expected variability of $R_{mod}$ and $\sigma$? That information would help put the retrieval errors into context better, but of course the lack of a long-term, global data record of these parameters is part of the motivation for this work!*

Unfortunately, we do not have any realistic assumptions for the changes in $R_{mod}$ and $\sigma$, there are just very rough approximations taken from Deshler (2008). These approximations were assumed to be constant the whole profile. Since with this study we intend to test the algorithm sensitivity, we don't think, that the chosen scenarios of $R_{mod}$ and $\sigma$ will bias the error assessments. The clarification has been provided in the revised manuscript.

*Sect. 4, 6th paragraph:*
*This paragraph notes changes in the particle distribution function shapes (a heavier "tail", for example) as the Manam aerosol is dispersed. Are all of these distributions single-mode log-normal? And if so, how can these apparent variations in shape occur except through variations of $\sigma$ with time? The last sentence of this paragraph says that $\sigma$ does not change significantly.*

For the whole study we assume unimodal log-normal distribution. By the "distribution width" in the last sentence of the paragraph we meant absolute distribution width ($w$),

defined in Eq. (2) of the manuscript, and not $\sigma$. Parameter $\sigma$ can change, though, as we mention in the Sect. 2, $\sigma$ is relative parameter and is not well suitable to describe the real width. The variations in the shape for Manam eruption occur through variation of both, $R_{mod}$ and $\sigma$, although this variations occur in a fashion, that $w$ is not changing drastically.

To reduce the possible confusion we have revised the manuscript and added "$w$" in the places, where we discuss the absolute distribution width.

*Sect. 5, 2nd paragraph:*
*The Damadeo et al. (2013) reference that describes the V7 SAGE II data refers back to Thomason et al. (2008) to describe their method of computing effective radius. That latter reference mentions log-normal size distributions, but it isn't totally clear that they assume single-mode log-normal distribution when computing the effective radius. This may be relevant as one compares the V7 SAGE II effective radius database to the values derived in this study.*
*Thomason et al. (2008) also mentions that the SAGE measurements lack sensitivity to aerosols with radius < 100 nm (0.1 um). This might also affect the results of the comparison presented in this section, since the median radius is near 0.1 um for many of the size distributions considered.*

That is true, Thomason et al. (2008) do not mention the the shape of the distribution of stratospheric aerosols, although in Sect. 4.2 Damadeo et al. (2013) the single mode log-normal distribution is assumed for the retrieval process. Similarly, in the earlier studies on SAGE II like Thomason et al. (1997) there is a link to Yue et al., (1986), where the unimodal distribution is also specifically mentioned. For that reason we think, that comparison is consistent. SCIAMACHY and SAGE II overlap period was volcanically quiscent, so following Yue (1999) the distribution will be very well described as unimodal.

The low sensitivity of SAGE II might have affected the comparison, and we have added this statement in the revised manuscript as one of the possible reasons for the differences.

*Sect. 5, 5th paragraph:*
*The final sentence summarizes the results of this comparison by calling 30% differences "a good agreement." What standard was used to assess how good this level of agreement is? Does it flow clearly from the estimated error in the SAGE effective radius, in combination with the stated precision of the SCIAMACHY aerosol property error estimates?*

Based on the reported errors, we think, that difference within 30% with SAGE II is a good result. Also, as reviewer mentions in previous comment, in Thomason et al. (2008) the low sensitivity to the particles with radius <0.1 $\mu$m is emphasized. Period from 2002 till 2005 is considered to be volcanically quiscent, with a large amount of smaller particles, thus SAGE II data can contain larger uncertainty, than for the volcanically active period. For that reason, we think that the results are quite good. Important to remember, there are not so many collocations with SAGE II, so the result could be biased and to give a more precise comparison.
We added the above mentioned reasons to the revised manuscript.

*Sect. 6:*
*The effective radius comparison is summarized in the last sentence, noting that agreement improves from $\approx 30\%$ at the bottom of the stratospheric aerosol layer to better than $10\%$ near its top. Might this be related to the fact that the aerosol distribution (as derived by Deshler et al.) is frequently bi-modal at lower altitudes, before becoming a single-mode distribution (as assumed in this study) at higher altitudes?*
*This leads to a more general question: Were any studies done to define how this algo-*

*rithm might perform if the true aerosol size distribution is not single-mode log-normal, but instead has some other (plausible) shape? Or if the true aerosol size distribution varies with altitude?*

The different shape in the lower and upper altitudes should not have influenced the comparison, because both SCIAMACHY and SAGE II assume unimodal log-normal particle size distribution, and the overlap period is considered to be volcanically quiscent, when the coarse mode is quite weak.

We have not conducted any studies in the algorithm performance on the bimodal distribution, but even Deshler et al. (2003) use unimodal distribution in some cases (not just at the upper altitudes). In multiple publications on the aerosol retrievals from space borne instruments unimodal distribution is considered to be quite representative. We will consider to provide the tests suggested by reviewer in the future studies.

*References:*
*Damadeo, R., Zawodny, J., Thomason, L., and Iyer, N.: SAGE version 7.0 algorithm: application to SAGE II, Atmospheric Measurement Techniques, 6, 3539–3561, 2013.*
*Deshler, T.: A review of global stratospheric aerosol: Measurements, importance, life cycle, and local stratospheric aerosol, Atmospheric Research, 90, 223–232, 2008.*
*Deshler, T., Hervig, M., Hofmann, D., Rosen, J., and Liley, J.: Thirty years of in situstratospheric aerosol size distribution measurements from Laramie, Wyoming (41 N), using balloon-borne instruments, Journal of Geophysical Research: Atmospheres, 108, 2003.*
*Rault, D. and R.P. Loughman, The OMPS Limb Profiler Environmental Data Record Algorithm Theoretical Basis Document and Expected Performance, IEEE Transactions on Geoscience and Remote Sensing, 51, doi:10.1109/TGRS.2012.2213093, 2013.*
*Rodgers, C. D.: Inverse methods for atmospheric sounding: theory and practice, vol. 2, World scientific, 2000.*

[Figure]

*Thomason, L. W., Burton, S. P., Luo, B.-P., and Peter, T.: SAGE II measurements of stratospheric aerosol properties at non-volcanic levels, Atmos. Chem. Phys., 8, 983-995, https://doi.org/10.5194/acp-8-983-2008, 2008.*

*von Savigny, C., Ernst, F., Rozanov, A., Hommel, R., Eichmann, K.-U., Rozanov, V., Burrows, J., and Thomason, L.: Improved stratospheric aerosol extinction profiles from SCIAMACHY: validation and sample results, Atmospheric Measurement Techniques,8, 5223–5235, 2015.*

Thomason, L. W., L. R. Poole, and T. Deshler. "A global climatology of stratospheric aerosol surface area density deduced from Stratospheric Aerosol and Gas Experiment II measurements: 1984–1994." Journal of Geophysical Research: Atmospheres 102.D7 (1997): 8967-8976.

Yue, G. K., M.P. McCormick, and W. P. Chu, Retrieval of composition and size distribution of stratospheric aerosols with the SAGE II satellite experiment, J. Atmos. Oceanic Technol., 3, 372-280, 1986.

Yue, G. K.: A new approach to retrieval of aerosol size distributions and integral properties from SAGE II aerosol extinction spectra, J. Geophys. Res., 104, 27 491–27 506, 1999.

---

## Referee Report (RR1)

**Referee Report on:**

**Aerosol particle size distribution in the stratosphere retrieved from SCIAMACHY limb measurements**

Elizaveta Malinina, Alexei Rozanov, Vladimir Rozanov, Patricia Liebing, Heinrich Bovensmann, and John P. Burrows

**General Comments:**

The explanations and modifications of the manuscript brought by the authors greatly improve the quality of the paper and address the main issues in a satisfactory way. Also the quality of the English language is greatly improved.

I still have a few minor issues and questions I think worth to be addressed.

**Specific comments:**

2. Instrument and applied algorithm:

p.6, l.29-30: "The particle number density (…) at 35 km". I suggest slightly modifying the sentence to make it fully clear that this is the model you considered, and no observation. E.g.: "The particle number density, N, is assumed to decrease (…)".

My previous comment on p.6, l.25: Above 35 km, it is known that all $H_2SO_4$ is only present in gas phase. It is irrelevant to retrieve sulfate aerosol above this altitude.

*Authors' response: Although there are not so many data above 35 km, that is true, that $H_2SO_4$ is strongly decreasing at higher altitudes and is presented just in the gas phase. For that reason we retrieve aerosol particle size distribution parameters from 18 to 35 km. However, it is known that there is some aerosol above 35 km (e.g. meteoritic dust), but in very small concentrations. It is taken into consideration by our aerosol number density profile, which is about 0.5 cm−3 at 35 km and is decreasing exponentially with increasing altitude. We define the aerosol profile above 35 km and below 18 km as sulfate aerosol in order to avoid jumps and unreasonable values at the lowermost and the uppermost retrieval altitudes. In Sect. 3.2 we show, that the trustworthy altitude range is 18-32 km.*

This response is, in my opinion, only half satisfactory. (1) The choice of the aerosol number density takes indeed into account the presence of particles above 35 km, but the difference in composition (mainly meteoritic dust, unlike the sulfate aerosol dominating at lower altitudes) is not taken into account in the size distribution retrieval. (2) Even if the authors are able to show afterward that neglecting the difference in aerosol composition has no consequence on the final result or even may help stabilizing the solution, I find problematic that the difference in composition is just ignored, what looks like propagating wrong information in a scientific journal (although the authors admittedly use the expression "*are assumed* to be sulfate droplets"). I think there is no problem to discuss this point in the paper using the response given in the author's reply, saying that despite the difference in aerosol composition below and above 35 km, a

similar composition is assumed to simplify the processing and to stabilize the solution, and that they show later in the paper that this is without consequence on the validity range of the solution.

3. Sensitivity studies:

General comment: I understand now the authors' rationale, and why my remark about the typical variability of N of several orders of magnitude was not relevant. I think that, to be fully clear and convincing, the authors might consider writing at the beginning of the section that they are assessing the sensitivity of the solution found at a given location. It looks obvious, but it seems I missed that point, and the other referee seems also to have some problem with this study.

Then, maybe a very clear way to approach the problem would be to look in the literature (Bingen et al.; Deshler et al., etc.) what is a maximal range for the variability of the 3 parameters (i.e. large value with respect to the typical uncertainty found for any ground-based and space-based technique). If the variation used in the present paper for the different sets of parameter is large with respect to this typical range assessed from the literature, I think the rationale followed by the authors will be fully convincing.

The explanation given to Referee 2 using the expression of the scattered light intensity is also quite useful to illustrate the reason of smaller sensitivity of the size distribution with respect to N. I suggest adding it in the text.

I also wonder if this study leads to satisfactory results for cases of volcanic aerosols. The variability observed during the ENVISAT period was up to a factor ~8 for the extinction (thus probably on the same order for N) in the UTLS, see e.g. Kremser et al., 2016 (op. cit.). Obviously, such cases are particularly interesting and probably the most wanted (cf. the importance of tropical eruptions during the period covered here). Hence, I think it is relevant to add some comment on such case – even if the authors have to state that the study is not valid in such a case.

p.11, l.4: Is the significant increase of the deviation in w up to 40% in the "volcanic (2N)" scenario related to the fact that a background size distribution is assumed in the calculation of the forward model, making it less appropriate to describe a volcanic case ?

p.11, l.21: See my remark 2 paragraphs above.

My previous comment: p.8, L. 13: "increase the uncertainty for the volcanic periods": this is certainly not the right way to do: the mean value remains the most probable. I believe this might be a reason, together with the assumptions of background aerosols made for the forward model, why SCIAMACHY's results show a systematic negative bias with respect to SAGE. (See Ch. 5).

*Author's response: We disagree with the reviewer on this point. First, as was shown in the manuscript, the $R_{mod}$ errors in the small scenarios are around 0.01 μm (relative error 10-20%). Implementation of the mean profile would noticeably increase uncertainty for the background cases, while for the volcanic cases, where the relative error is about 20-25%, the uncertainty will remain. Second, there are no known assessments of changes in the particle number density profiles during the Envisat operating period, all the known*

*assumptions were based on SAGE II climatology, which included the colossal eruption of Pinatubo, which is not representative for 2002-2012.*

*(...)*

I am afraid there must be a misunderstanding. Is the method followed by the authors as follows:

1. Choice of a baseline scenario => gives the "true values"; e.g.:
   a. "small" => {$R_{mod}$=0.06 µm; σ=1.7, N following the exponential vertical profile described in p.6, ll 29-30};
   b. "volcanic (2N)" => {$R_{mod}$=0.20 µm; σ=1.2, [N following the exponential vertical profile described in p.6, ll 29-30]x2}
2. Choice of a set of perturbed scenarios; e.g.:
   a. "small" => {$R_{mod}$=0.06 µm+Gaussian noise; σ=1.7+Gaussian noise, N following the exponential vertical profile described in p.6, ll 29-30};
   b. "volcanic (2N)" => {$R_{mod}$=0.20 µm+Gaussian noise; σ=1.2+Gaussian noise, [N following the exponential vertical profile described in p.6, ll 29-30]x2+Gaussian noise})
3. For these perturbed scenarios: computation of the corresponding limb radiance
4. For the computed "perturbed" limb radiance, retrieval of $R_{mod}$ and σ.
5. Comparison of all retrieved "perturbed" {$R_{mod}$, σ}with the initial set chosen for the baseline scenario in point 1.

?

If this is what the authors are doing, then I don't know why they call the 3$^{rd}$ scenario "unperturbed" nor why the "true values" of {$R_{mod}$, σ} have such a behaviour in the case "volcanic (2N)" in figs. 4-6, but I discard my comment on the case "volcanic (2N)".

On the contrary, if, in the case "volcanic (2N)":

- the baseline scenario is {$R_{mod}$=0.20 µm; σ=1.2, N following the exponential vertical profile described in p.6, ll 29-30},
- the perturbed scenarios are {$R_{mod}$=0.20 µm+Gaussian noise; σ=1.2+Gaussian noise, [N following the exponential vertical profile described in p.6, ll 29-30]x2+Gaussian noise},

then I maintain my criticism, because the Gaussian distribution of all perturbations should be centred on the baseline scenario, which represent the "truth" you want to retrieve.

The authors should clarify the method used in the paper.

4. Results and discussion:

p.13, l. 14: I know I used the expression "individual profiles" (vs. "monthly profiles") in my previous referee report, but I am not sure that, here, out of the context of my previous remark, this expression is clear of any reader. The authors might consider being more explicit, by using for instance an expression such as "profiles retrieved from individual measurements".

5. Comparison with SAGE II:

p.15, ll. 4 and 14-16: Considering the time series depicted in Fig. 15, especially at 18.0 km, I must say I don't find that the "tendency to show the same features" is striking, nor that differences around 30% (depending on the reference chosen, it might be estimated to more than 50% in some cases) look particularly a good performance, since this difference exceeds from far the variability found over the whole 3-year period. Therefore, I suggest removing the part of the sentence about the tendency to show the same features in l.4, and the highly subjective estimation of "good result": they weaken the author's argumentation and do not bring much to it. The discussion concerning the clear bias and its interpretation is far more interesting.

6. Conclusions:

p.15, l.24: Please qualify in "for the considered unperturbed N profile".

Additional comment for this section: As discussed above, in the forward model used to retrieve the extinction from limb scattering sounders, the assumption of background aerosols is used, and it is not clear to me in which extend this assumption influences the particle size retrieval in case of volcanic aerosols. This point should be at least mentioned, even if the answer to this question is unknown or uncertain.

---

## Author Response (AR2)

**Author's response to the Referee comments on the revised manuscript "Aerosol particle size distribution in the stratosphere retrieved from SCIAMACHY limb measurements" by Elizaveta Malinina et al.**

We thank the reviewers for the time they spent reading and commenting the manuscript the second time. We answered the reviewer's questions and provided the explanations, which were needed. To distinguish the referee's comments from the author's responses, the comments are shown in italicized font and the responses are highlighted in blue.

**General Comments:**

*The explanations and modifications of the manuscript brought by the authors greatly improve the quality of the paper and address the main issues in a satisfactory way. Also the quality of the English language is greatly improved.*
*I still have a few minor issues and questions I think worth to be addressed.*

**Specific Comments:**

2. Instrument and applied algorithm:

*p.6, l.29-30: "The particle number density (…) at 35 km".I suggest slightly modifying the sentence to make it fully clear that this is the model you considered, and no observation. E.g.: "The particle number density, N, is assumed to decrease (…)".*

The text has been revised in accordance with the reviewer's comment.

*My previous comment on p.6, l.25: Above 35 km, it is known that all H2SO4 is only present in gas phase. It is irrelevant to retrieve sulfate aerosol above this altitude.*
*Authors' response: Although there are not so many data above 35 km, that is true, that $H_2SO_4$ is strongly decreasing at higher altitudes and is presented just in the gas phase. For that reason we retrieve aerosol particle size distribution parameters from 18 to 35 km. However, it is known that there is some aerosol above 35 km (e.g. meteoritic dust), but in very small concentrations. It is taken into consideration by our aerosol number density profile, which is about $0.5$ $cm^{-3}$ at 35 km and is decreasing exponentially with increasing altitude. We define the aerosol profile above 35 km and below 18 km as sulfate aerosol in order to avoid jumps and unreasonable values at the lowermost and the uppermost retrieval altitudes. In Sect. 3.2 we show, that the trustworthy altitude range is 18-32 km.*
*This response is, in my opinion, only half satisfactory. (1) The choice of the aerosol number density takes indeed into account the presence of particles above 35 km, but the difference in composition (mainly meteoritic dust, unlike the sulfate aerosol dominating at lower altitudes) is not taken into account in the size distribution retrieval. (2) Even if the authors are able to show afterward that neglecting the difference in aerosol composition has no consequence on the final result or even may help stabilizing the solution, I find problematic that the difference in composition is just ignored, what looks like propagating wrong information in a scientific journal (although the authors admittedly use the expression "*are assumed* to be sulfate droplets"). I think there is no problem to discuss this point in the paper using the response given in the author's reply, saying that despite the difference in aerosol composition below and above 35 km, a similar composition is assumed to simplify the processing and to stabilize the solution, and that they show later in the paper that this is without consequence on the validity range of the solution.*

The clarification has been added to the revised manuscript.

3. Sensitivity studies:

*General comment: I understand now the authors' rationale, and why my remark about the typical variability of N of several orders of magnitude was not relevant. I think that, to be fully clear and convincing, the authors might consider writing at the beginning of the section that they are assessing the sensitivity of the solution found at a given location. It looks obvious, but it seems I missed that point, and the other referee seems also to have some problem with this study.*

In the revised manuscript we highlighted, that the analysis has been done at the particular altitude.

*Then, maybe a very clear way to approach the problem would be to look in the literature (Bingen et al.; Deshler et al., etc.) what is a maximal range for the variability of the 3 parameters (i.e. large value with respect to the typical uncertainty found for any ground-based and space-based technique). If the variation used in the present paper for the different sets of parameter is large with respect to this typical range assessed from the literature, I think the rationale followed by the authors will be fully convincing.*

In this study the goal was to assess the relative sensitivity of the observed limb radiance to the variations in diffrent parameters. As mentioned in the early replies and in the manuscript, there are no known published assessments of the variations of $R_{mod}$ and $\sigma$ at the period of interest (2002-2012) for the tropical region. Nevertheless, we believe, that the showed range of parameters is roughly representative based on the data of OPCs (Deshler at al, 2003). In Figs.1-3 the data for the fine mode $R_{mod}$, $\sigma$ and the sum of the fine and coarse mode $N$ from OPCs measurements in Wyoming from August 2002 till May 2012 are presented. According to the colorbars, $R_{mod}$ was varying from around 0.02 to 0.16 $\mu$m, $\sigma$ from around 1.2 to 1.8 and $N$ was roughly changing within a factor of 2 from its background values. Thus, the range of the variatiopns considered for the study was roughly comparable with the real amplitudes of $R_{mod}$, $\sigma$ and $N$. We added this statement to the revised manuscript.

*The explanation given to Referee 2 using the expression of the scattered light intensity is also quite useful to illustrate the reason of smaller sensitivity of the size distribution with respect to N. I suggest adding it in the text.*

The explanation has been added to the revised manuscript.

*I also wonder if this study leads to satisfactory results for cases of volcanic aerosols. The variability observed during the ENVISAT period was up to a factor ∼8 for the extinction (thus probably on the same order for N) in the UTLS, see e.g. Kremser et al., 2016 (op. cit.). Obviously, such cases are particularly interesting and probably the most wanted (cf. the importance of tropical eruptions during the period covered here). Hence, I think it is relevant to add some comment on such case – even if the authors have to state that the study is not valid in such a case.*

Even if the variability of the extinction coefficient was up to a factor ∼8, it does not necessarily mean that particle number density was varying by the same factor. As the aerosol extinction coefficient is dependent on both, aerosol extinction cross section $\alpha_{aer}$ and number density $N$: $Ext = \alpha_{aer}(R_{mod}, \sigma)N$, all three parameters contribute into the changes in aerosol extinction. While $Ext$ is linearly proportional to $N$, the dependency on $R_{mod}$ and $\sigma$ is non-linear and rather complicated. To illustrate this, we present in Fig. 4 aerosol extinction coefficient profile for the following distributions: $R_{mod}$=0.08 $\mu$m, $\sigma$=1.6 and background $N$ profile, $R_{mod}$=0.08 $\mu$m, $\sigma$=1.6 and doubled $N$ profile and $R_{mod}$=0.16 $\mu$m, $\sigma$=1.3 and background $N$ profile. Analyzing Fig. 4

it can be seen, that an increase of $N$ by a factor of 2 results in an increase in $Ext$ by the same factor, while a change of $R_{mod}$ by a factor of 2 (even with the decreased $\sigma$) increases $Ext$ by a factor of 3. Thus, changes in the aerosol extinction coefficient do not necessarily result from the changes in $N$ of the same magnitude.

We are fully convinced, that our data is representative within the reported errors. As discussed above, the OPCs data for the sum of the fine and coarse mode $N$ presented in Fig. 3, shows that the assumption about $N$ varying within a factor of 2 during the SCIAMACHY operating period is rather reasonable.

For sure, the assumption of the background $N$ profile introduces some errors for the volcanic cases. Although, as it was shown in Fig. 1 of the manuscript, in the volcanic cases the sensitivity to changes in both, $R_{mod}$ and $\sigma$, is higher compared to the background conditions. To eliminate the uncertainty related to the assumption on the $N$ profile, some additional information needs to be included to constrain the retrieval. This, however, is a subject of the upcoming studies.

*p.11, l.4: Is the significant increase of the deviation in w up to 40% in the "volcanic (2N)" scenario related to the fact that a background size distribution is assumed in the calculation of the forward model making it less appropriate to describe a volcanic case?*

Yes, the relative error in $w$ of 40% in the "volcanic (2N)" case is related to the assumed background profile and intesified by the small absolute value of $w$. Currently, we see no reliable way to adjust the assumption on $N$ based on the measurements. As it was described in the previous round of the review, we believe, that the background profile is more optimal approach.

*p.11, l.21: See my remark 2 paragraphs above.*

Please, see the answer to that paragraph.

*My previous comment: p.8, L. 13: "increase the uncertainty for the volcanic periods": this is certainly not the right way to do: the mean value remains the most probable. I believe this might be a reason, together with the assumptions of background aerosols made for the forward model, why SCIAMACHY's results show a systematic negative bias with respect to SAGE. (See Ch. 5).*

*Author's response:* We disagree with the reviewer on this point. First, as was shown in the manuscript, the $R_{mod}$ errors in the small scenarios are around 0.01 $\mu m$ (relative error 10-20%). Implementation of the mean profile would noticeably increase uncertainty for the background cases, while for the volcanic cases, where the relative error is about 20-25%, the uncertainty will remain. Second, there are no known assessments of changes in the particle number density profiles during the Envisat operating period, all the known assumptions were based on SAGE II climatology, which included the colossal eruption of Pinatubo, which is not representative for 2002-2012.

*(...)*

*I am afraid there must be a misunderstanding. Is the method followed by the authors as follows:*

1. *Choice of a baseline scenario =>gives the "true values"; e.g.:*

    (a) *"small" =>{$R_{mod}$=0.06 $\mu m$; $\sigma$=1.7, N following the exponential vertical profile described in p.6, ll 29-30};*

    (b) *"volcanic (2N)" =>{$R_{mod}$=0.20 $\mu m$; $\sigma$=1.2, [N following the exponential vertical profile described in p.6, ll 29-30]×2}*

2. *Choice of a set of perturbed scenarios; e.g.:*

(a) "small" =>{$R_{mod}$=0.06 μm+Gaussian noise; σ=1.7+Gaussian noise, N following the exponential vertical profile described in p.6, ll 29-30};

(b) "volcanic (2N)" =>{$R_{mod}$=0.20 μm+Gaussian noise; σ=1.2+Gaussian noise, [N following the exponential vertical profile described in p.6, ll 29-30]×2+Gaussian noise})

3. For these perturbed scenarios: computation of the corresponding limb radiance

4. For the computed "perturbed" limb radiance, retrieval of $R_{mod}$ and σ.

5. Comparison of all retrieved "perturbed" {$R_{mod}$, σ} with the initial set chosen for the baseline scenario in point 1.

?

If this is what the authors are doing, then I don't know why they call the 3rd scenario "unperturbed" nor why the "true values" of {$R_{mod}$, σ} have such a behaviour in the case "volcanic (2N)" in figs. 4-6, but I discard my comment on the case "volcanic (2N)".
On the contrary, if, in the case "volcanic (2N)":

- the baseline scenario is {$R_{mod}$=0.20 μm; σ=1.2, N following the exponential vertical profile described in p.6, ll 29-30},

- the perturbed scenarios are {$R_{mod}$=0.20 μm+Gaussian noise; σ=1.2+Gaussian noise, [N following the exponential vertical profile described in p.6, ll 29-30]×2+Gaussian noise}

then I maintain my criticism, because the Gaussian distribution of all perturbations should be centred on the baseline scenario, which represent the "truth" you want to retrieve.
The authors should clarify the method used in the paper.

We believe that there must be a misunderstanding, as none of the approaches described by the reviewer is applied.
The algorithm works as follows:

1. The intensities are simulated for the scenarios described in the Tab. 1 of the manuscript. For "volcanic (2N)" scenario N profile described in p.6, ll 29-30 was multiplied by a factor of 2 from 12 to 23 km. The values from Tab. 1 considered to be "true", and these are the values, that we expect to retrieve.

2. The Gaussian noise is added to the simulated intensities assuming the signal-to-noise ratios estimated from the measurements. To ensure a proper statistics 100 independent noise sequences are generated for each simulated limb radiance. For that reason we call the scenario with $R_{mod}$=0.11 μm and σ=1.37 "unperturbed", as in this scenario none of the particle size distribution parameters is perturbed.

3. For all the simulated limb radiances, $R_{mod}$ and σ are retrieved, including the "unperturbed" scenario.

4. All retrieved $R_{mod}$ and σ are compared to the corresponding "true" values.

We have revised the manuscript to avoid possible confusion.

_4. Results and discussion:_
_p.13, l. 14: I know I used the expression "individual profiles" (vs. "monthly profiles") in my previous referee report, but I am not sure that, here, out of the context of my previous remark,_

*this expression is clear of any reader. The authors might consider being more explicit, by using for instance an expression such as "profiles retrieved from individual measurements".*

The sentence has been corrected.

*5. Comparison with SAGE II:*
*p.15, ll. 4 and 14-16: Considering the time series depicted in Fig. 15, especially at 18.0 km, I must say I don't find that the "tendency to show the same features" is striking, nor that differences around 30% (depending on the reference chosen, it might be estimated to more than 50% in some cases) look particularly a good performance, since this difference exceeds from far the variability found over the whole 3-year period. Therefore, I suggest removing the part of the sentence about the tendency to show the same features in l.4, and the highly subjective estimation of "good result": they weaken the author's argumentation and do not bring much to it. The discussion concerning the clear bias and its interpretation is far more interesting.*

The phrase "tend to show the same features" and the last paragraph of the section have been removed. Instead we added the statement, that the difference between SCIAMACHY and SAGE II is not time dependent.

*6. Conclusions:*
*p.15, l.24: Please qualify in "for the considered unperturbed N profile".*

The sentence has been revised.

*Additional comment for this section: As discussed above, in the forward model used to retrieve the extinction from limb scattering sounders, the assumption of background aerosols is used, and it is not clear to me in which extend this assumption influences the particle size retrieval in case of volcanic aerosols. This point should be at least mentioned, even if the answer to this question is unknown or uncertain.*

There might have been a misunderstanding about the applied algorithm. Unlike the algorithms used to retrieve the particle size distribution parameters from the occultation instruments (e.g. Bingen et al., 2004; Thomason et al., 2008), our algorithm does not include the intermediate step of the aerosol extinction retrieval, and $R_{mod}$ and $\sigma$ are retrieved directly from the radiances. To avoid the possible confusion we added this clarification to the Sec. 2.3 "Algorithm description" of the revised manuscript.

Acknowledgments: The OPCs data was downloaded from
http://www-das.uwyo.edu/~deshler/Data/Aer_Meas_Wy_read_me.htm

[Figure]

Figure 1: Mode radius (fine mode) in the period from August 2002 till May 2012 from OPC measurements in Wyoming.

[Figure]

Figure 2: Sigma ($\sigma$) (fine mode) in the period from August 2002 till May 2012 from OPC measurements in Wyoming.

[Figure]

Figure 3: Particle number density (sum of the fine and coarse mode) in the period from August 2002 till May 2012 from OPC measurements in Wyoming.

[Figure]

Figure 4: Aerosol extinction coefficient at 750 nm for the ditsributions with $R_{mod}$=0.08 $\mu$m, $\sigma$=1.6 and background $N$ profile (blue solid line); $R_{mod}$=0.08 $\mu$m, $\sigma$=1.6 and doubled $N$ profile (blue dashed line) and $R_{mod}$=0.16 $\mu$m, $\sigma$=1.3 and background $N$ profile (red solid line).

[revised manuscript text omitted]